# Changes in gene expression predictably shift and switch genetic interactions

Xianghua Li [1], Jasna Lalić[1], Pablo Baeza-Centurion [1], Riddhiman Dhar [1] & Ben Lehner [1,2,3]

Non-additive interactions between mutations occur extensively and also change across conditions, making genetic prediction a difficult challenge. To better understand the plasticity of genetic interactions (epistasis), we combine mutations in a single protein performing a single function (a transcriptional repressor inhibiting a target gene). Even in this minimal system, genetic interactions switch from positive (suppressive) to negative (enhancing) as the expression of the gene changes. These seemingly complicated changes can be predicted using a mathematical model that propagates the effects of mutations on protein folding to the cellular phenotype. More generally, changes in gene expression should be expected to alter the effects of mutations and how they interact whenever the relationship between expression and a phenotype is nonlinear, which is the case for most genes. These results have important implications for understanding genotype-phenotype maps and illustrate how changes in genetic interactions can often—but not always—be predicted by hierarchical mechanistic models.

[1] Centre for Genomic Regulation (CRG), The Barcelona Institute of Science and Technology, Dr. Aiguader 88, Barcelona 08003, Spain. [2] Universitat Pompeu Fabra (UPF), Barcelona, Spain. [3] ICREA, Pg. Luis Companys 23, Barcelona 08010, Spain. Correspondence and requests for materials should be addressed to B.L. (email: ben.lehner@crg.eu)

To interpret personal genomes, make accurate genetic predictions and understand evolution we need to be able to predict the effects of mutations and also to understand how they combine (interact). Large-scale projects[1] and deep mutagenesis[2–9] have revealed that mutations frequently interact non-additively, which makes accurate genetic prediction a difficult challenge[10].

Non-additive genetic (epistatic) interactions between gene deletions and loss-of-function alleles have been mapped genome-wide in budding yeast, revealing that both pairwise[1] and higher order[11,12] epistasis are widespread. Similarly, epistasis is frequent when combining all possible pairs of mutations between two different proteins[2], between natural genetic variants[13,14] and between mutations selected during adaptation to new environments[15,16]. Systematic mutagenesis of individual proteins[4–9] and RNAs[3,17,18] has also revealed widespread epistasis within individual macromolecules.

Comparisons across species[19–22], conditions[23,24], time[25] and cell types[26,27], have repeatedly found that genetic interactions are plastic, changing in different cells and conditions. This plasticity has important implications for both evolution and genetic disease. For example, a synthetic lethal genetic interaction between a cancer-causing mutation and a drug or gene inhibition that will kill one cell often proves ineffective in other cells that carry the same driver mutation[27].

Why do both the effects of mutations and genetic interactions change across conditions, cell types and species? Comparing between any two cell types, environmental conditions or species, there are typically thousands of molecular differences such as changes in gene expression, making this a difficult question to answer. We reasoned that one way to address this question would be to ask it in a minimal system in which we could quantify the effects of mutations and how mutations interact and then test how these effects and interactions change in response to a simple perturbation of the cellular state: a change in the expression of the mutated gene.

There is a rich theoretical literature on how both biochemistry and regulatory networks can generate many of the classic statistical phenomena of genetics[28–30], including interactions between mutations[29,31,32]. For example, the thermodynamics of protein folding[31] and molecular interactions[2] result in non-linear relationships between changes in free energy and the activity of individual molecules and complexes. Similarly, regulatory systems often have steep sigmoidal dose-response functions because of cooperativity, molecular titration and feedback[30,33]. The kinetic coupling of enzymes can also generate non-linear

expression–fitness functions[34]. Thus, pioneering theoretical work has shown that many mechanistic aspects of molecular biology are expected to produce non-additive genetic interactions between mutations[32].

The phage lambda repressor (CI) is one of the best characterized proteins, serving as a paradigm for both gene regulation[35] and quantitative biology[36–39]. The detailed and quantitative understanding of how this protein functions makes it an ideal system in which to address our question of how mutational effects and the interactions between mutations change when a system is perturbed. Previously, it was shown that both the direction of mutational effects and genetic interactions between two mutations in a promoter targetd by CI can change depending upon whether the repressor is expressed or not. Modelling suggested that the cause of this is mutations pleiotropically affecting binding of both the repressor and RNA polymerase[37].

Here we show that, even in this minimal system, the effects of individual mutations and the interactions between mutations in a protein change extensively when the expression level of the protein is altered. Indeed we show that even a simple perturbation can result in the interactions between mutations changing in direction, flipping between positive (suppressive) and negative (enhancing) epistasis. We show that these seemingly complicated changes can be both understood and predicted using a mathematical model that propagates the effects of mutations on protein folding to the cellular phenotype. More generally, we show that changes in gene expression will alter the effects of mutations and how they interact whenever the relationship between expression and a phenotype is nonlinear. Given that this is the case for most genes, shifts and switches in the interactions between mutations should be widely expected when the expression level of a gene changes.

## Results

**Mutagenesis of the lambda repressor at two expression levels.**
We used doped oligonucleotide synthesis to introduce random mutations into the 59-amino acid helix-turn-helix DNA-binding domain of CI, and quantified the ability of each genotype to repress expression of a fluorescent protein (GFP) from the $P_R$ (Promoter R) target promoter by fluorescence-activated cell sorting into near neutral (Output1) and partially detrimental (Output2) bins and deep sequencing (Fig. 1a–c, Supplementary Fig. 1). From the sequencing read counts, we calculated enrichment scores for each variant in each bin. We then estimated the

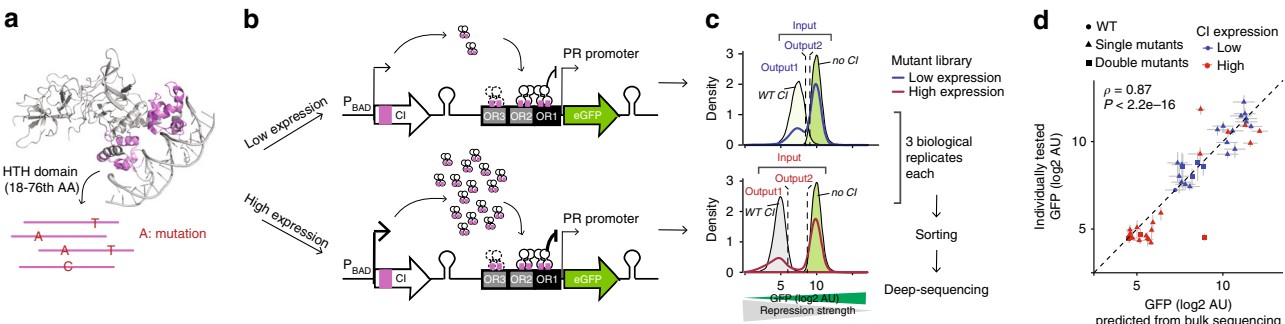

**Fig. 1** Deep mutagenesis of CI DNA-binding domain at two expression levels. **a** CI-operator complex with mutated HTH domain in magenta. **b** Experimental design. **c** Distribution of GFP target gene expression when the mutant library is expressed at high and low levels and when WT or no CI is expressed. Sorted populations with GFP level similar to the wild-type population (Output1), and population with intermediate GFP level (Output2) were collected for deep sequencing. **d** Correlation of target gene expression estimated by deep sequencing with target gene expression individually quantified for wild type, 22 single and double mutants at low and high expression levels. Error bars denote standard error of the mean from four (y-axis) and three (x-axis) biological replicates

GFP expression using the enrichment scores from both bins (Fig. 1c, Supplementary Fig. 1, see Methods). CI was expressed at a level similar to that observed in phage lysogens[35] (see Methods). We quantified both the effects of single mutants and the genetic interactions between pairs of mutations. We then repeated the experiment expressing CI at a higher expression level and re-quantified the mutation effects and genetic interactions. The effects of wild type, 18 single and four double mutants when measured individually were highly correlated with their effects quantified in the pooled assay by deep sequencing at both expression levels (Fig. 1d, rho = 0.87, $P < 2e-16$, $n = 46$; rho = 0.82 and rho = 0.71, respectively, for low and high CI expression conditions, $n = 23$, see also Supplementary Fig. 2).

At both expression levels, the single (Fig. 2a, $n = 351$) and double amino acid-change mutants (Fig. 2b, $n = 468$) had a bimodal distribution of target gene expression levels, with the low and high modes centred on the phenotypes observed for synonymous and premature stop codon-containing genotypes, respectively (Fig. 2a, b). These bimodal distributions of mutational effects are consistent with observations for many different proteins[2,5–8,40–42], as is the shifted distribution of double mutant phenotypes towards higher expression of the target gene (i.e. reduced activity[2,8]) (Fig. 2a, b). Also consistent with previous deep mutagenesis datasets[6–8,43], mutations in the core residues of the protein were more detrimental (reduced repression of the target gene) than mutations in solvent-exposed residues (Fig. 2c,

Supplementary Fig. 3). Mutations in residues contacting DNA were also more detrimental than mutations in solvent-exposed residues (Fig. 2c, Supplementary Fig. 3). As expected, mutations to less similar amino acids were also more detrimental, as were mutations predicted to reduce the free energy of protein folding or DNA-binding (Supplementary Fig. 3c–f). Mutations to less hydrophobic amino acids were detrimental in the core and mutations that introduced a negative charge were detrimental at positions that contact DNA (Supplementary Fig. 2g–j).

**Mutational effects in CI change non-linearly.** Comparing the expression of the target gene when the same single (Fig. 2d) or double (Fig. 2e) mutant genotypes were expressed at high and low levels revealed a nonlinear relationship, with four main classes of genotypes: (1) genotypes with little effect at either high or low expression (~42% of single mutants), (2) genotypes having little effect at high expression but detrimental effects at low expression (~26% of single mutants), (3) genotypes that are partially detrimental at high expression but behave similarly to null alleles at low expression (~5% of single mutants) and (4) genotypes that behave similarly to null alleles at both expression levels (~20% of single mutants); 7% of mutants, including mutations partially detrimental at both expression levels, did not fall into any of these four classes. This unmasking of detrimental mutation effects at low expression levels has been previously observed for mutations

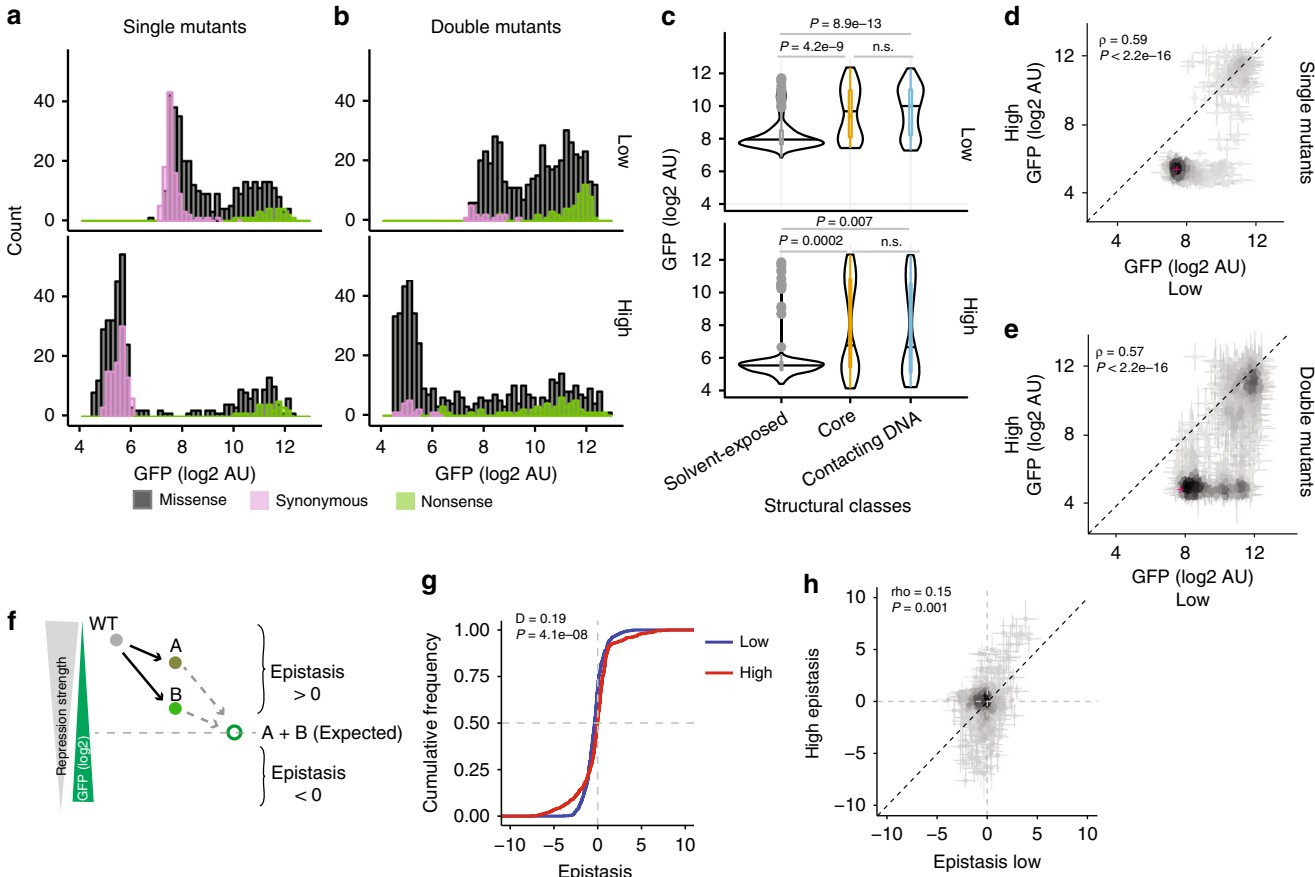

**Fig. 2** Comparison of mutational effects and genetic interactions at two expression levels. **a, b** Histogram of the mean mutational effects of single ($n = 351$) (**a**) and double ($n = 468$) (**b**) missense amino acid variants together with synonymous ($n = 114$ for single, $n = 37$ for double) and nonsense ($n = 21$ for single, $n = 47$ for double) variants. **c** Effects of single mutants in different structural regions. Classes compared using Kruskal-Wallis test with post hoc Dunn's test. **d, e** Comparison of mean mutational effects at the two expression levels. **f** Log-additive definition of epistasis. **g** Cumulative distributions of mean epistasis scores at the two expression levels ($n = 468$). Distributions compared using two-sample Kolmogorov–Smirnov test. **h** Mean epistasis scores at the two expression levels. Error bars in **d**, **e**, **h** denote standard error of the mean

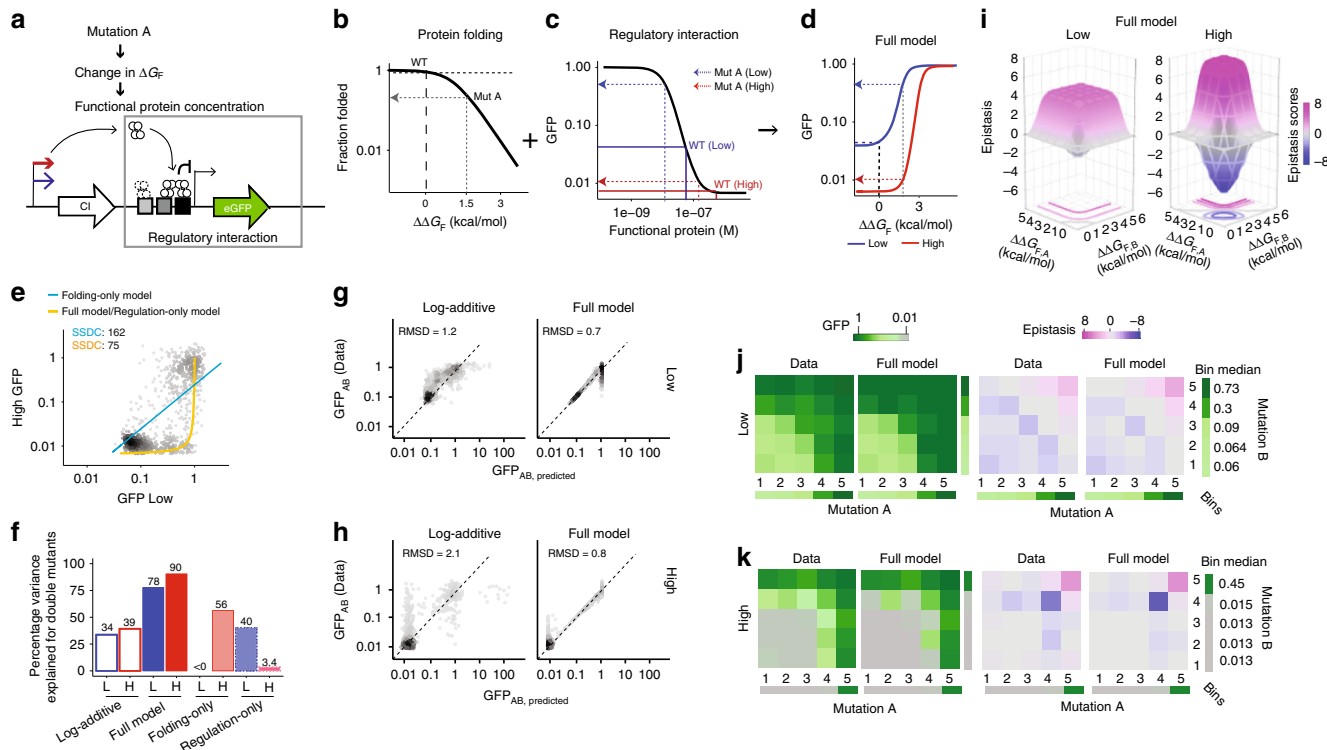

**Fig. 3** Combined model of protein folding and regulatory interaction. **a** Mutations alter the free energy of protein folding ($\Delta G_F$) and so protein concentration and repression of the target gene. **b–d** Relationships between change in folding energy ($\Delta\Delta G_F$) and the fraction of folded protein (**b**), protein concentration and target gene expression (**c**), and change in folding energy ($\Delta\Delta G_F$) and target gene expression at low (blue) and high (red) expression (**d**). The effect of an example mutation (**a**) is indicated on each graph. **e** Regulatory interaction-only but not the protein folding-only model predicts the inverse relationship between target gene expression at the two expression levels. SSDC: sum of the squared distance from the curve. **f** Percentage of variance explained for double mutations for each model. 'L' indicates low expression and 'H' indicates high expression of CI. **g**, **h** Observed vs. predicted target gene expression for the log-additive and full folding + regulation model at low (**g**) and high (**h**) CI expression. RMSD: root-mean-square-deviation between the predicted and observed data. **i** Predicted epistasis at low and high expression for the full model. **j**, **k** Model-predicted and experimentally-observed target gene expression and epistasis when combining mutants at low (**j**) and high (**k**) expression. Mutations were ordered by their effects into five equally populated bins and the median target gene expression and epistasis plotted for each bin combination

in a region of yeast Hsp90[41] and also for human disease-causing variants[44].

**Changing expression alters how mutations in CI interact.** We quantified epistasis between pairs of mutations as the difference between the observed and expected phenotypes based on a log additive model[45]. A positive epistatic interaction means that repression of the target gene by the double mutant is greater than expected and a negative interaction means that it is less than expected (Fig. 2f). The distribution of epistasis scores differed between the two expression levels of the protein, with more strong positive and negative interactions at high expression (Fig. 2g, two-sample Kolmogorov–Smirnov Test $P = 4.1e-8$, $D = 0.19$, $n = 468$). Furthermore, epistasis scores of the same pairs of mutations at the two protein expression levels correlated only weakly (Fig. 2h, rho $= 0.15$, $P = 0.001$, $n = 468$). Plotting epistasis against the expected double mutational effects revealed systematic trends in the data (Supplementary Fig. 4). As expected, double mutants with high expected target gene expression tended to interact positively at both low and high expression. On the other hand, double mutants with intermediate expected outcomes had stronger negative interactions at low expression, and double mutants with low expected target gene expression had stronger negative interactions at high expression (Supplementary Fig. 4).

**A mathematical model predicts changes in epistasis.** What accounts for these systematic patterns of epistasis and also their

dependence on expression level? To address this, we turned to a previously published quantitative model of repression of the $P_R$ promoter by CI[36] (Supplementary Fig. 5a). Briefly, the model describes the probability of CI repressing the expression of the target gene as a function of CI concentration (Fig. 3a, c). We first mapped each single mutant's effect from the target gene expression level to the concentration of active CI. We then extended this model to include the effects of mutations on the folding of CI and estimated changes in the free energy of folding for each single mutant (see Methods). To predict the CI concentration and the resulting expression of the GFP target gene for each double mutant, we summed the change in free energy for each single mutant and then mapped the total free energy to a change in protein folding and concentration, which was in turn mapped to altered repression of the target gene (Supplementary Fig. 5b, c). We compared the behaviour of the full model (Fig. 3b–e) to that of models that only considered protein folding (Supplementary Fig. 5d) or repression of the target gene by CI (Supplementary Fig. 5e).

Both the full model and the transcription regulation-only model correctly predict the shape of the relationship between mutational effects at low and high expression (Fig. 3e, $n = 819$). However, only the full model provides good prediction of the phenotypes of double mutants from the phenotypes of the single mutants (Fig. 3f). The full model (Fig. 3g, h), but not the folding-only or regulation-only models (Supplementary Fig. 6), also captures the systematic trends in how mutations combine at both low and high expression.

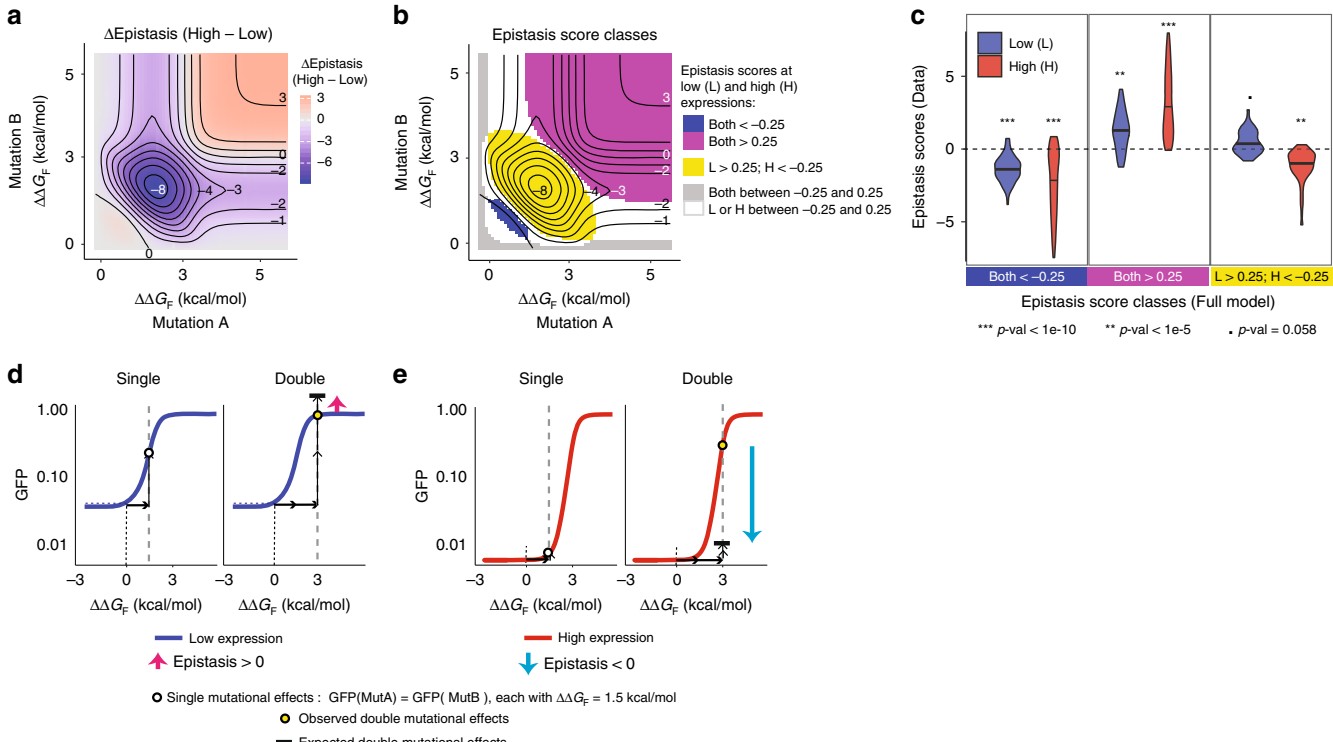

**Fig. 4** Changes in expression alter both the strength and sign of epistasis. **a, b** Changes in epistasis strength (**a**) and class (**b**) between low and high expression predicted by the model. **c** Experimentally determined epistasis scores for double mutants with the indicated model-predicted epistasis scores. The one-sample Wilcoxon signed rank test was performed to test whether average epistasis scores are significantly different from 0. *P*-values are adjusted with Bonferroni multiple test correction method. **d, e** The same pair of mutations can interact positively at low expression (**d**) and negatively at high expression (**e**)

**The cause of expression-dependent epistasis.** Inspection of the model reveals that it is the nonlinear S-shaped relationship[46] between protein concentration and target gene repression that causes the concentration-dependence of both mutational effects and genetic interactions (Fig. 3d, i). Each mutation has a fixed effect on the free energy of protein folding (Fig. 3b). When combining two mutations, the changes in free energy are summed and so alter the fraction of folded protein according to the nonlinear relationship in Fig. 3b. However, because the relationship between protein concentration and target gene expression follows a nonlinear S-shaped curve, the same change in protein concentration can lead to a different change in target gene expression depending upon the starting protein concentration (Fig. 3b–d). The S-shaped relationship between protein concentration and target gene expression therefore transforms the concentration-independent effects of mutations on protein folding (Fig. 3b) into concentration-dependent changes in target gene expression (Fig. 3c, d), resulting in concentration-dependent epistasis (Fig. 3i–k, Supplementary Fig. 7).

**Changes in gene expression reverse the sign of epistasis.** Comparing how mutations combine at different expression levels in the full model revealed that changes in expression not only alter the magnitude of genetic interactions but can also switch their direction of interaction (between positive and negative interactions, Fig. 4a, b). Re-analysis of the experimental data validated this prediction, with mutations in the regime predicted by the model switching from positive to negative epistasis as the expression level increased (Fig. 4c, Supplementary Fig. 8). In other words, genetic interactions that are suppressive at one expression level can become enhancing at another expression level (Fig. 4d, e). It is worth noting that these changes are

different from sign epistasis which refers to the mutational effects themselves switching from positive to negative in different genetic backgrounds[47]. We did not observe sign epistasis in our experiment or model. Our model and data therefore show that changes in expression can alter both the strength and the type of epistasis between mutations.

**Changes in gene expression alter epistasis for many genes.** To what extent should we expect these conclusions to apply to other genes? Mutational effects and genetic interactions will be expression-level dependent whenever the relationship between expression and a phenotype is nonlinear. Such nonlinear expression-fitness functions are indeed very common in biology, because of the abundance of cooperation, competition, and feedback, with nonlinear functions used to model almost all aspects of cell biology[48]. Moreover, the relationship between expression level and fitness (growth rate) has been systematically quantified for 81 yeast genes and, for all genes sensitive to a change in expression in the tested conditions, the expression-fitness function was nonlinear, with the three most frequent expression-fitness functions being a concave increase in fitness as expression increases, a concave decrease, or a concave peaked function[49].

We quantified epistasis and its sensitivity to concentration changes in these three common expression-fitness functions of yeast genes[49]. For many yeast genes, fitness increases as a concave function as their expression is increased from zero to a fitness plateau close to the wild-type expression level (Fig. 5a). For these genes, epistasis changes in magnitude but not sign as the expression level changes (Fig. 5c, e, g, Supplementary Fig. 9). Similar results are seen for genes where fitness decreases as a concave function as expression is increased (Supplementary Fig. 9).

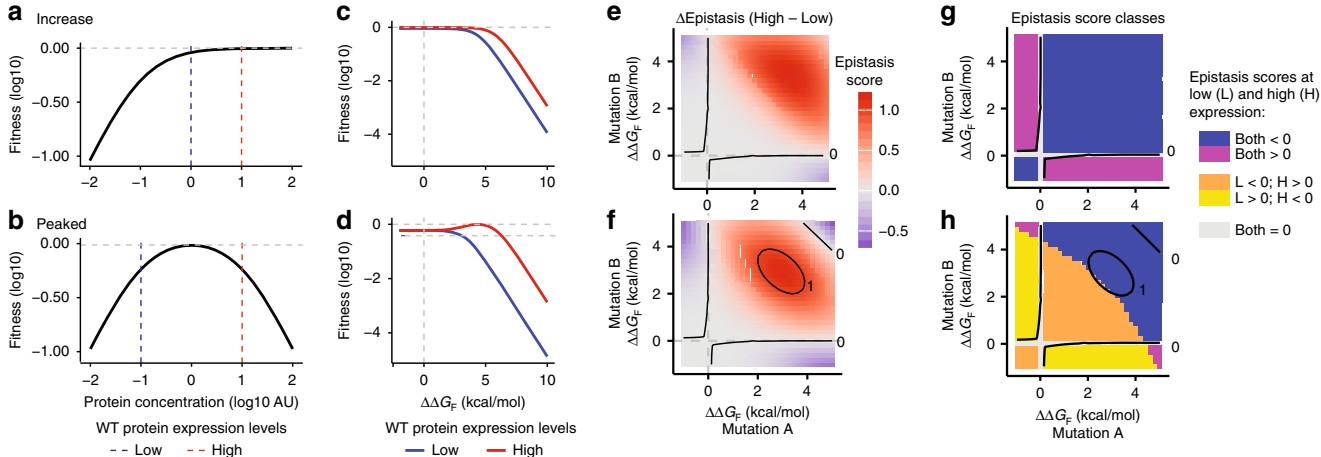

**Fig. 5** Expression-fitness functions generate concentration-dependent genetic interactions. **a**, **b** Two common expression-fitness functions in budding yeast. **c**, **d** Relationship between change in free energy of protein folding and fitness for these functions. **e**, **f** change in epistasis magnitude between high and low expression. **g**, **h** Change in epistasis sign between high and low expression

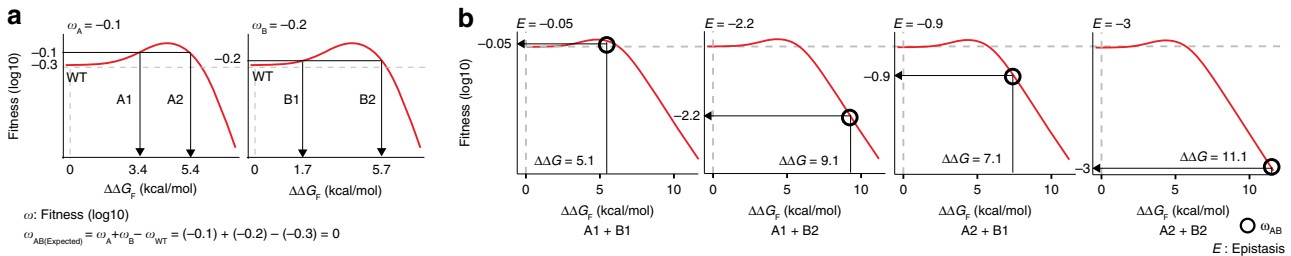

**Fig. 6** Unpredictable double mutant phenotypes. **a** For 'peaked' expression-fitness functions such as that shown in Fig. 5b, the same change in fitness can be caused by two different changes in folding free energy. **b** For a pair of single mutant phenotypes there can therefore be up to four possible double mutant outcomes

Multiple genes in yeast have a peaked expression-fitness landscape[49]. For these genes epistasis can change substantially and also switch in direction as the expression level changes because of the non-monotonic relationship between the free energy of protein folding and fitness (Fig. 5b, d, f, h).

**Ambiguous genetic predictions**. Finally, analysing how mutations combine in genes with different expression-fitness functions we realised that for some genes accurate predictions for how mutations combine will never be possible from two fitness measurements alone, even with a perfect mechanistic model. Specifically, when there is a non-monotonic relationship between the expression level and a phenotype, the same observed phenotype for a single mutant can map to two or more different free energies of protein folding, leading to multiple possible double mutant phenotype predictions for each mutation pair (Fig. 6, Supplementary Fig. 10). For these genes, even a perfect mechanistic model is therefore insufficient to predict how mutations of precisely measured effects combine to alter a phenotype. In such cases it will always be necessary to make additional measurements —for example of intermediate phenotypes, such as protein concentrations—to predict how two mutations will combine to alter a phenotype.

## Discussion
Non-additive interactions between mutations complicate genotype-phenotype maps and so make genetic prediction a difficult challenge. Further, the interactions between mutations

can also change across conditions, for example between different types of cancer[26,27]. Using the very well-characterised minimal system of the lambda repressor, we found that a change in expression altered both the effects of individual mutations and how these mutations combined. Moreover, we found that changes in expression altered both the strength and the type of interactions between mutations, with interactions switching from positive (suppressive) to negative (enhancing) at different expression levels.

In our analyses we only considered the effects of mutations that alter the free energy of protein folding. Altered protein stability is likely to be the most common effect of amino acid changing mutations[31]. However, subsets of mutations will have additional effects, for example altering the affinity and kinetics of molecular interactions. In future work it will be important to study how mutations with different molecular effects interact with each other, as well as with mutations that affect stability and with changes in expression. Our model also makes the assumption that the effects of mutations on protein stability are independent of the expression level but this may sometimes not be the case, for example because of chaperone titration[50] or interactions with other molecules[2,51]. Concentration-dependent changes in the effects of mutations on protein stability will lead to further shifts in mutational effects and genetic interactions as a gene's expression changes.

Although our experimental work focussed on the lambda repressor, by analysing other common expression-fitness functions, we have shown that our conclusions are likely to apply to many genes. Indeed changes in expression will transform the

effects of mutations and their interactions whenever the relationship between expression and a phenotype is nonlinear. In yeast, where expression-fitness functions have been systematically quantified[49], this is normally the case: for most genes the growth rate of the organism does not depend in a linear way on the gene's expression level. For many genes, therefore, changes in expression alone will drive changes in mutational effects and genetic interactions. Thus we should expect that genetic interactions will change extensively across conditions and cell types in an animal, as well as between individuals in a population and between different species. Analyses of genetic interactions across conditions[23,24,52], time[25], cell types[26,27], and species[19–22] are highly consistent with this.

Changes in genetic interactions are relevant to both agriculture[53] and human genetic disease. For example, variable epistasis may contribute to the tissue-specificity of human disease mutations as well as the cancer type-specificity of interactions between cancer driver mutations[26]. Moreover, the success of synthetic lethal strategies to specifically kill target cells depends on the stability of these interactions. Many examples now exist of synthetic lethal gene perturbations that are effective in one cancer or cell type but ineffective in other cell types, and the most successful targets will be interactions that are very stable across individuals and perturbations[26,27]. Furthermore, the plasticity of epistasis caused by changes in expression level suggests that the accessible and most likely evolutionary paths will change over time as the expression level of a gene is altered.

Importantly, we found that the seemingly complicated shifts and switches in genetic interactions as the expression level of the lambda repressor changed could be understood and predicted using a hierarchical mechanistic model that propagates the effects of mutations on the free energy of protein folding to the cellular phenotype. Considering just the effects of mutations on protein folding or just how the repressor regulates gene expression could not account for the changes in interactions. We envisage that such multi-step models that propagate the effects of mutations on protein stability to higher-level phenotypes may prove generally useful for genetic prediction and for understanding how mutations combine to alter phenotypes, including in human disease. Together with additional work[54], this highlights the importance of multi-scale models in biology. In particular, although there are many models for how biochemical parameters influence higher-level cellular and organ phenotypes, these models rarely connect to genetic variation. Deep mutagenesis of additional biological processes, including those with more complex dynamical behaviour, will provide a more complete view of how mutations impact on phenotypes and fitness.

Finally, although we found that a hierarchical model provided accurate genetic prediction for the lambda repressor, we also realised that there are cases where such a mechanistic model will fail to accurately predict how mutations combine to alter phenotypes. Specifically, when there is a non-monotonic relationship between the concentration of a protein and a phenotype, it is sometimes not possible to predict how two mutations will combine, even with a detailed mechanistic model. This is because some phenotypes map to two or more possible changes in protein concentration and so to multiple changes in the free energy of protein folding. Without additional measurements it is not possible to tell which of the underlying changes is causing the phenotype. This results in multiple possible outcomes when mutations of known phenotypic effect are combined. In these cases, additional measurements such as of intermediate phenotypes such as protein concentrations will always be required for accurate genetic prediction.

## Methods

**Microbe strain and growth conditions.** *E.coli* BW27783 MK01 strain (kindly provided by the M.Isalan lab), modified to homogenously express arabinose-induced genes[55] was used to express the mutant library. A single colony of the *E. coli* BW27783 MK01 strain was picked from Luria-Bertani (LB) agar plate, grown overnight at LB liquid medium supplemented with chloramphenicol to 14 µg/per ml concentration at 37 °C. The 500 µl overnight growth media with cells mixed with 500 µl of 50% glycerol were stored at −80 °C freezer. For experiments, cells were always grown at 37 °C in LB liquid medium supplemented with appropriate antibiotics. For specific experimental growth conditions, please refer to the method details in the following section.

**Mutant oligonucleotide library synthesis and amplification.** A 250-nucleotide-long oligonucleotide library was synthesized by TriLink BioTechnologies. Library oligonucleotides contain a 177-nucleotide-long sequence of the CI Helix-Turn-Helix domain (52th–210th nucleotide bases, based on CI ORF GeneID:3827059), doped at each position with 0.4% of each of the three non-reference nucleotides. The doped region is flanked by invariant sequences corresponding to the wild type sequences of immediate upstream (36 nucleotide bases) and downstream (37 nucleotide bases) of the doped region and used as constant overhang regions for the PCR primers to bind. The designed oligonucleotide sequence is:

5′- *CCATTAACACAAGAGCAGCTTGAGGACGCACGTCGCcttaaagcaattta tgaaaaaaagaaaaatgaacttggcttatcccaggaatctgtcgcagacaagatggggatggggcagtcaggcgttggtg ctttatttaatggcatcaatgcattaaatgcttataacgccgcattgcttgcaaaaattctcaaagttagcgttgaagaatt- tAGCCCTTCAATCGCCAGAGAAATCTACGAGATGTATG* 3′-

Upper case indicates the constant regions and lower case the doped sequence.

The doped library was dissolved in 500 µl MilliQ water as a stock solution, and 10 µl of the stock solution was further diluted in 500 µl of MilliQ water as a working solution. The working solution oligonucleotide concentration was estimated to be 390 ng per µl based on NanoDrop (Thermofisher Scientific) measurement of ssDNA concentration. Next, the working solution 'doped' library was further diluted by a factor of 100, and a total of about 40 ng was used as the template to synthesize the complementary strand as well as to be amplified. Polymerase chain reaction (PCR) was performed using Phusion high fidelity PCR kit (Thermo Scientific) with primers that bind to the constant regions of the doped library oligonucleotide (Supplementary Table 1). Each 50 µl PCR reaction consisted of 10 µl of the doped library oligonucleotide as the template, 10 µl of 5X Phusion HF reaction buffer, 1 µl of 10 mM dNTP (NEB), 2.5 µl of 10 µM forward and reverse primers each, 0.5 µl of Phusion polymerase and 12.5 µl MilliQ. PCR reactions followed the manufacturer's instruction for a standard protocol. Eighteen PCR cycles were performed to minimize incorporation of PCR errors to the library. PCRs were performed with annealing temperature at 55 °C and extension at 72 °C for 30 s. The fragment with the correct size (230 nucleotide bases) was visualized and retrieved using the 2% size-select E-gel purification system (Invitrogen). To achieve optimal ligation efficiency, the size-selected PCR fragment was further purified with the MiniElute PCR purification kit (QIAGEN) to remove excess salt. The Gibson assembly (GA) system was used to ligate the PCR fragments to the modified plasmid backbone (see below) following the standard GA protocol.

**Plasmid constructs.** The CI open reading frame (GeneID:3827059) was cloned into the bacterial expression vector pBADM-11 (obtained from CRG biomolecular screening & protein technologies unit), between the arabinose-inducible promoter pBAD and three stop codons in all three reading frames (tagttaagtga), followed by the strong synthetic bidirectional terminator L3S2P21[56]. The $P_R$ promoter (that overlaps with OR3, OR2 and OR1 repressor binding sites) followed by the RBS-GFP (LVA) ORF[57] was cloned downstream of the L3S2P21 terminator. Three stop codons in all three reading frames (tagttaagtga) were cloned immediately downstream of the GFP ORF and upstream of the pBADM-11 intrinsic rrnB_T2 terminator.

The two plasmid constructs (pCIPR plasmids) used in our experiments—the construct expressing CI to a high concentration (pCIPR-High) and the construct expressing CI to a low concentration (pCIPR-Low)—differed in the DNA sequences between the predicted strongest ribosome binding sequence (RBS) and the ATG start codon of the CI gene. In pCIPR-High, the start codon is immediately after the RBS. In pCIPR-Low the start codon is 82 nucleotides downstream of the RBS.

The pCIPR-High and pCIPR-Low plasmids were linearized by removing the coding region of the CI helix-turn-helix motif (HTH) domain that contains the doped sequence. The doped oligonucleotide library and the linearized plasmids were assembled using the GA system (master mix provided by CRG biomolecular screening & protein technologies unit) following the standard protocol. The assembly reactions were dialysed using 0.025µm VSWP membrane filters (Merk Millipore Ltd) and electroporated into the high efficiency commercial NEB10β competent cells (NEB, C3020K). After recovery in 500 µl Super Optimal broth with Catabolite repression (SOC) culture media at 37 °C for one hour, an aliquot of the cells was plated on Luria-Bertani (LB) agar plate with 100 µg/ml ampicillin to examine the transformation efficiency, and the rest was diluted 1 in 200 in fresh Luria-Bertani (LB) broth with 100 µg per ml ampicillin for overnight growth. About 780,000 independent transformant colonies were obtained for the mutant

plasmid library construction. Plasmids were purified using the Qiagen Midiprep kit (cat.12143) and the purified plasmids were then used as the mutant plasmid library.

**Making highly efficient electro-competent cells.** We chose the *E.coli* BW27783 MK01 strain (kindly provided by the Isalan lab)[55], modified to homogenously express arabinose-induced genes, to express the mutant library. A single chloramphenicol-resistant colony of was picked into 4 ml LB medium with 2.8 μl of 20 mg per ml chloramphenicol and let grow for 3.5 h at 37 °C. In all, 2 ml of this pre-culture bacterial media was then diluted into 250 ml of pre-warmed 2 ×Ty media with 175 μl 20 mg per ml chloramphenicol for 2 h and 10 min and ensured that the OD600 did not exceed 0.6. The culture was cooled down on ice for 5 minutes, divided into four 50 ml falcon tubes and centrifuged at a speed of 17.8×g for 5 min at 4 °C. The cell pellets were suspended in 50 ml cold Milli-Q water in each of the four falcon tubes and then centrifuged again at a speed of 17.8×g for 5 min at 4 °C. After that, the cell pellets were suspended in 50 ml cold Milli-Q water in two falcon tubes, and the centrifugation step was repeated as before. A final wash of cell pellets was performed in cold 10% glycerol. After centrifuging for 7 min at 4 °C and 17.8×g, the supernatant was shaken away and the cells were re-suspended in their own juice.

**Sorting cells based on CI mutants' phenotypes.** In all, 0.5 μl of 200 ng per μl pCIPR plasmids were transformed into 25 μl electrocompetent cells made on the same day, inside a 0.1cm-gap cuvette (Bio-Rad) using the Gene Pulser Xcell™ electroporation system (Bio-Rad), with the pre-set protocol for *E.coli* transformation. Cells were recovered in SOC culture media at 37 °C for 1 h, and an aliquot of the cells was plated on LB agar plate with 100 μg per ml ampicillin to examine the transformation efficiency. One transformation with this step produced millions of transformants without creating a bottleneck. Cells were grown overnight in 25 ml LB medium with 100 μg per ml ampicillin. An aliquot of the overnight culture was diluted 1 in 100 into LB media containing 100 μg per ml ampicillin, 0.4% glucose and 0.2% arabinose and grown at 37 °C for 2.5 h to reach an OD600 of about 0.7. The bacteria culture was further diluted in 5 with fresh medium (same composition) and the cells grown for another hour, after which the OD600 was about 0.9. As a control for no CI induction (no repression of the target gene GFP), cells were grown in the LB medium without arabinose but with glucose. All experiments included cells with plasmids containing the wild type CI genotype (positive control) and cells containing an empty pBADM-11 plasmid (to quantify cell autofluorescence) in addition to the cells carrying the mutant library. After the induction of CI expression, cells were immediately diluted 1 in 500 into Phosphate buffered saline (PBS) and put on ice before FACS.

Sorting was performed at the CRG FACS core facility. A FACSAria II SORP sorter along with the FACSDiva Version 6.1.2 software was used to sort the cells. Bacterial cells were selected based on side scatter (SSC) and forward scatter (FSC), and gate selection was based on FITC-A fluorescence filter for GFP (Supplementary Fig. 11a). Cells were sorted into three gates: the near neutral gate was defined as including 90% of the matching wild type population. The completely detrimental gate included 90% of non-repressed high GFP population (no CI induction). The intermediate population between the two populations mentioned above (about 3~4% of all the library population was in this gate) was also collected (Supplementary Fig. 12b). Purity of sorting was examined by passing the sorted cells through the FACS again immediately after sorting, and recording the population proportions belonging to the sorted gate. At least 30 million cells were sorted per biological replicate. Cells from the completely detrimental gate were not further processed for deep sequencing, for the following two reasons: (1) Variants from the detrimental gate were expected to be enriched with insertions or deletions, and stop codons that we do not want in our analysis; (2) amino acid substitutions that are completely detrimental (therefore enriched in the completely detrimental gate) can be deduced based on variants' frequency in the input, near neutral fraction, and intermediate fraction, thus we decided not to sequence the most detrimental gate fraction to be cost-effective.

Sorted cells were kept on ice in PBS in 15 ml falcon tube each. They were centrifuged at 17.8×g at 4 °C for 30 min. The supernatant was removed carefully, and the plasmid-prep was performed directly form the cell pellets. Plasmids from the sorted cells (together with the unsorted input cells) were extracted immediately with the QIAprep Spin Miniprep kit (QIAGEN). The mutagenized region was amplified using barcoded PCR primers (Supplementary Table 1) for 25 cycles using hot start Phusion polymerase (Thermo Scientific) in 50 μl reactions, following the manufacturer instruction. PCR products were purified using the E-gel 2% size-select system (Invitrogen) to remove smaller fragments. In order to produce three full biological replicates, the procedure described up to this—from transformation of the mutation plasmid library to cell sorting and plasmid extraction—was performed three times on three different days (Supplementary Fig. 11c).

Concentration of each purified PCR product was measured on NanoDrop (Thermofisher Scientific). Equimolar quantities of three independent amplifications of the input library (Input) and equimolar quantities of three output replicates from near neutral population (Output1) were pooled together in one Eppendorf tube (Sample1). Equimolar quantities of three output replicates from intermediate population (Output2) were pooled together as a separate sample in a different Eppendorf tube (Sample2). The two samples were sent to EMBL Genomics Core Facility where two PCR-free sequencing libraries were prepared

and sequenced on Illumina HiSeq2000 platform. The PCR-free sequencing library Sample1 was run on two lane of an Illumina HiSeq 2500 for each CI concentration experiment. The PCR-free sequencing library Sample2 was multiplexed with other samples to about 10% of one lane loading, considering the small size of the cell population.

**Verification of mutational effects.** 22 genotypes (Supplementary Fig. 2c, d and Supplementary Table 2) were selected based on their enrichment scores at both CI concentrations for re-testing in order to cover a wide phenotypic space. In this reference set, we included all mutation types including synonymous, nonsense, missense and also some double mutations. Individual genotypes were constructed using the NEB Q5 site-directed mutagenesis kit (NEB cat. E0554S) with the wild type pCIPR-High and pCIPR-Low plasmids as templates. After verifying the sequences by Sanger sequencing, we picked four colonies from each genotype to examine their target gene GFP expression levels (Supplementary Fig. 2c, d). The experiment was performed in one batch on the same day so that the results from this experiment could be used as a confirmation set to which other FACS experiment sets can be mapped. LSR Fortesta florescence analyser was used at the CRG FACS Core facility.

GFP signal and the shape information of 10,000 cells per biological replicate were recorded, and the.FCS files from the recordings were analyzed using the FlowCore package in R. Cells were filtered based on SSC and FSC, and the first 3000 cell recordings were discarded to avoid cross-well contamination. The mean output GFP signal (in AU, arbitrary units) from about 5000 cells in each biological replicate of individual variant was calculated after the filtering process. The mean GFP signal and standard error of the mean for each variant were obtained from each biological replicate.

In order to verify estimated GFP expression levels converted from the enrichment scores based on the reference set of 22 genotypes (see below Data analysis section), we selected 9 additional genotypes (Supplementary Fig. 2, Supplementary Table 3) after mapping the enrichment scores to the target gene GFP expression levels. The experiment procedure was the same for the 22 genotypes mentioned above.

**Quantification of CI protein expression.** The relative amount of CI protein at the two expression levels was quantified by tagging CI with GFP at its C-terminus with the flexible linker amino acid sequence GSAGSAAGSGEF[58]. The PR-GFP sequences were removed from the original pCIPR-High and pCIPR-Low plasmids to make plasmids pCIGFP-High and pCIGFP-low. Fluorescence from CI-induced cells was analysed using a LSR Fortesta florescence analyser at the CRG FACS Core facility (Supplementary Fig. 12a). In the same experiment, GFP calibration beads (CloneTech) were used to calibrate and obtain exact molecule numbers based on the GFP signal (Supplementary Fig. 12b, c). For quantification, mean GFP signals and standard errors of were calculated from four biological replicates.

**From sequencing data to target gene expression.** Our data analysis pipeline consists of three main parts: (1) Filtering. (2) Mapping enrichment scores to the target gene (GFP) expression levels. (3) Correcting for the batch effects (Supplementary Fig. 11c) and the detection limits set by the experiment. The processed final datasets for the analysis were organised both on nucleotide level and amino acid level. Even though our conclusions were mainly based on amino-acid level mutational effects, the dataset with nucleotide-level mutational effects was needed as reference.

The analyses from sequencing data to GFP expression level were all performed on the nucleotide-level, and the amino-acid level mutational effects were examined based on the processed nucleotide-level datasets. Whenever involving combining replicates (at the level of enrichment scores, predicted GFP singles at the nucleotide level and at amino acid level), the random error model was used.

**From Illumina sequencing reads to variant counts.** To extract variant counts from the raw sequencing data, we adapted the pipeline developed by our group in a previously published project[59]. Specifically, the raw sequencing data was demultiplexed with the SABRE software [https://github.com/najoshi/sabre] and paired reads were merged with the PEAR software[60] with parameters set not to allow any mismatches in the overlap regions. Reverse complementation of merged sequences was performed when necessary with the fastx_reverse_complement tool [http://hannonlab.cshl.edu/fastx_toolkit/]. Then the primer sequences were trimmed using the seqtk tool [https://github.com/lh3/seqtk]. Finally, the number of occurrences of each variant was counted with fastx_collapser [http://hannonlab.cshl.edu/fastx_toolkit/] and a custom python script[59].

**Calculating enrichment scores and filtering.** Variants up to 2-Hamming-distance nucleotide changes from the wild type sequence with at least 100 read counts in all three input replicates were selected for further analysis (Supplementary Figure 13a,b). The 100 read count threshold included all the 1-Hamming-distance nucleotide changes ($n = 531$) but only about 11% ($n = 10,862$ for low expression dataset) and 7% ($n = 3686$ for high expression dataset) of all the 2-Hamming-distance nucleotide changes observed. This restriction was necessary to obtain the confident variant counts. The threshold was chosen based on the logic

that each bacterial cell be expected to carry hundreds of plasmid copies (pUC replication origin). Considering experimental steps of plasmid extraction and PCR amplifications until obtaining read counts from Illumina sequencing, we reasoned that variants observed less than 100 read counts were likely to be from too few cells, resulting in unreliable enrichment scores for the following steps.

Enrichment scores for each variant $v$ from each experimental replicate $i$ (REPi), for each sorted cell output $j$ (Oj with O1 as near neutral fraction and O2 as partially detrimental fraction) were calculated as follows:

$$S_{v,Oj,REPi} = log2\left(\frac{C_{v,Oj,REPi} + 0.5}{C_{wt,Oj,REPi} + 0.5}\right) - log2\left(\frac{C_{v,input,REPi} + 0.5}{C_{wt,input,REPi} + 0.5}\right) \quad (1)$$

With $C$ as sequencing read counts, $v$ as variant, wt as wild type. A pseudo count of 0.5 was added to avoid log 0. Poisson-based error for each variant for each replicate for each output ($SE_{v,Oj,REPi}$) was also calculated using the formula below:

$$SE_{v,Oj,repi} = \sqrt{\frac{1}{C_{v,input,REPi} + 0.5} + \frac{1}{C_{wt,input,REPi} + 0.5} + \frac{1}{C_{v,Oj,REPi} + 0.5} + \frac{1}{C_{wt,Oj,REPi} + 0.5}} \quad (2)$$

In order to merge scores over replicates for each output and for each variant, and to be able to filter variants based on the standard errors of the mean, a random-effect error model as proposed by Rubin et al. for this type of data analysis[61] was used.

The details are as follows:

For the first iteration, for each output, an initial error $\widehat{SE}^2_{v,Oj,1}$ for each variant was calculated based on its standard deviation from the unweighted mean.

$$\hat{S}_{v,Oj,1} = \frac{\sum_{i=1}^{n=3} S_{v,Oj,REPi}}{3} \quad (3)$$

$$\widehat{SE}^2_{v,Oj,1} = \frac{1}{n-1} \times \sum_{i=1}^{3}\left(S_{v,Oj,REPi} - \hat{S}_{v,Oj,1}\right)^2 \quad (4)$$

The initial weighted mean enrichment score for each output was calculated as the follows:

$$\hat{S}_{v,Oj,1} = \frac{\sum_{i=1}^{3}\left(S_{v,Oj,REPi} \times \left(\widehat{SE}^2_{v,Oj,1} + SE^2_{v,Oj,REPi}\right)^{-1}\right)}{\sum_{i=1}^{3}\left(\widehat{SE}^2_{v,Oj,1} + SE^2_{v,Oj,REPi}\right)^{-1}} \quad (5)$$

For each iteration $k$, the standard error was calculated as follows:

$$\widehat{SE}^2_{v,Oj,k+1} = \widehat{SE}^2_{v,Oj,k} \times \frac{\sum_{i=1}^{3}\left(\widehat{SE}^2_{v,Oj,k} + SE^2_{v,Oj,REPi}\right)^{-2} \times \left(S_{v,Oj,REPi} - \hat{S}_{v,Oj,k}\right)^2}{\sum_{i=1}^{3}\left(\widehat{SE}^2_{v,Oj,k} + SE^2_{v,Oj,REPi}\right)^{-1} - \frac{\sum_{i=1}^{3}\left(\widehat{SE}^2_{v,Oj,k}+SE^2_{v,Oj,REPi}\right)^{-2}}{\sum_{i=1}^{3}\left(\widehat{SE}^2_{v,Oj,k}+SE^2_{v,Oj,REPi}\right)^{-1}}} \quad (6)$$

After 50 iterations ($k = 50$), the final mean enrichment score and standard error for each variant for each output were calculated as shown in Eqs. (7) and (8), respectively.

$$\hat{S}_{v,Oj} = \frac{\sum_{i=1}^{3}\left(S_{v,Oj,REPi} \times \left(\widehat{SE}^2_{v,Oj,50} + SE^2_{v,Oj,REPi}\right)^{-1}\right)}{\sum_{i=1}^{3}\left(\widehat{SE}^2_{v,Oj,50} + SE^2_{v,ROj,EPi}\right)^{-1}} \quad (7)$$

$$\widehat{SE}_{v,Oj} = \left(\sum_{i=1}^{3}\left(\widehat{SE}^2_{v,Oj,50} + SE^2_{v,Oj,REPi}\right)^{-1}\right)^{-0.5} \quad (8)$$

In order to estimate the overall errors of enrichment scores for each variant and to filter only the confident data for the following data analysis, the estimated errors from Output1 ($\widehat{SE}_{v,o1}$) and Output2 ($\widehat{SE}_{v,o2}$) were combined with the following formula:

$$\widehat{SE}_{v,o1+o2} = \left(\widehat{SE}^2_{v,o1} + \widehat{SE}^2_{v,o2}\right)^{0.5} \quad (9)$$

Variants with $\widehat{SE}_{v,o1+o2} > 1$ were removed for downstream analyses (Supplementary Fig. 13c, d).

**Mapping enrichment scores to GFP signal**. In order to calculate GFP signals from enrichment scores, we first examined the relationships between GFP signals and enrichment scores from individually assayed confirmation data set (Supplementary Table 2). As designed by the experiment, the smaller enrichment score from the Output1 $S_{v,o1}$ was, the higher GFP signal (more detrimental) of a variant was (Supplementary Fig. 14a). Enrichment scores from the Output2 $S_{v,o2}$ (the intermediate fraction) did not relate monotonically to the mean GFP signal, because variants enriched in Output2 ($S_{v,o2}$) were depleted for both strongly detrimental and near neutral variants (Supplementary Fig. 14b).

To examine the possibility of predicting GFP signals with a linear combination of the two enrichment scores for each replicate from each expression level experiment, we built linear models to predict the mean GFP signals with $S_{v,o1,REPi}$ and $S_{v,o2,REPi}$ with the confirmation dataset. The calculated GFP signal from the mean enrichment scores predicted the individual variants' GFP signals well (Supplementary Fig. 14f). However, the predictions were not completely linearly related with the observed GFP signals.

In order to improve the GFP signal predictions based on the enrichment scores, for each biological replicate, we transformed each $S_{v,o2,REPi}$ to $S_{v,o2,trans,REPi}$ based on its relationship with $S_{v,o1,REPi}$ such that variants predicted to be detrimental by $S_{v,o1,REPi}$ would have higher $S_{v,o2,trans,REPi}$ and variants predicted to be near neutral by $S_{v,o1,REPi}$ would have lower $S_{v,o2,trans,REPi}$ (Supplementary Fig. 14d, e).

The logic behind this transformation was as follows: 1) A potentially beneficial mutation was expected to be enriched in Output1 and depleted in Output2 ($S_{v,o1,REPi} > 0$ and $S_{v,o2,REPi} < 0$). We kept the Output2 score as it was. 2) An intermediately detrimental mutation was expected to be enriched in Output2 ($S_{v,o2,REPi} > 0$) regardless of its enrichment score in Output1. We kept its enrichment score in Output2 as it was as well. 3) A very detrimental mutation was expected to be depleted both in Output1 and Output2 ($S_{v,o1,REPi} < 0$ and $S_{v,o2,REPi} < 0$). In order to distinguish $S_{v,o2,REPi}$ of these variants from that of potentially beneficial mutations (the first case, where $S_{v,o2,REPi}$ is also smaller than 0), we transformed $S_{v,o2,REPi}$ to a positive value and bigger than the intermediately detrimental variants' $S_{v,o2,REPi}$. This way, a transformed $S_{v,o2,trans,REPi}$ was expected to be bigger for more detrimental mutations (Supplementary Figure 14c). To avoid influence by extreme outliers, 95th quartiles ($Q$) were used as thresholds for detrimental mutations ($S_{v,o1,REPi} < Q(S_{wt\_syn,o1,REPi}, 0.95)$) and as an approximate for the maximum $S_{v,o2,REPi}$ before transformation. To summarize, the equation follows:

$$S_{v,o2_{trans},REPi} = \begin{cases} if\left(S_{v,o1,REPi} < Q\left(S_{WT\_syn,o1,REPi}, 0.95\right) \& S_{v,o2,REPi} < 0\right), \\ Q\left(S_{o2,REPi}, 0.95\right) + abs\left(S_{v,o2,REPi}\right), \\ else, \\ S_{v,o2,REPi} \end{cases} \quad (10)$$

A linear model was built again to predict the mean GFP signals for each expression level experiment with the mean enrichment scores $\hat{S}_{v,o1}$ and $\hat{S}_{v02_{trans}}$ using the confirmation dataset. Inverse of the variance was used as weights. This linear model improved the prediction of GFP signal in the low CI expression dataset (Supplementary Fig. 14g). For the high expression dataset, the $\hat{S}_{v,o2_{trans}}$ coefficient was not significant (Supplementary Table 4, Supplementary Fig. 14g) and including the $S_{v,o2\_trans,REPi}$ did not improve logged GFP signal (as an output of mutational effects, denoted O) $O_{v,REPi}$ predictions (note $R^2$ and the median RMSD did not change in the predictions for high expression dataset, Supplementary Figure 14g). Therefore, we set $S_{v,o2\_trans,REPi} = 0$ when calculating signals and the errors for the high CI expression dataset in the following equations to avoid inflating the errors of the estimation (refer to Eq. (12)).

$$O_{v,REPi} = log_2\left(GFP_{v,REPi}\right) = \alpha + \beta \cdot S_{v,o1,REPi} + \gamma \cdot S_{v,o2_{trans},REPi} \quad (11)$$

$O_{v,REPi}$ above is the output GFP signal in log scale for each variant in each of the three biological replicates $i$ and the coefficients $\alpha$, $\beta$, $\gamma$ (Supplementary Table 4) derived from the linear model trained with the confirmation dataset.

A measurement error for the log GFP signal ($OE_{v,REPi}$) for each variant $v$ in each replicate $i$ and for each CI concentration (high and low) was calculated with the following formula:

$$OE_{v,REPi} = \sqrt{\begin{array}{l} \beta^2 \cdot SE^2_{v,o1,REPi} + \gamma^2 \cdot SE^2_{v,o2_{trans},REPi} \\ +2 \cdot \beta \cdot \gamma \cdot cov\left(S_{v,o1,REPi}, S_{v,o2_{trans},REPi}\right) \\ +\beta E^2 \cdot S^2_{v,o1,REPi} + \gamma E^2 \cdot S^2_{v,o2_{trans},REPi} + \alpha E^2 \end{array}} \quad (12)$$

Where $\beta E^2$, $\gamma E^2$, $\alpha E^2$ are squares of the standard errors of the estimated $\alpha$, $\beta$ and $\gamma$ coefficients respectively, and $cov(S_{v,o1,REPi}, S_{v,o2,trans,REPi})$ is the covariance between $S_{v,o1}$ and $S_{v,o2,trans}$ for each replicate (Supplementary Table 5).

**Correcting technical biases**. Each biological replicate from FACS sorting on different days had different ranges of GFP expression levels (GFP index, Supplementary Fig. 11c) and these biases were reflected on the estimated $O_{v,REPi}$ (Supplementary Fig. 15a, b). In order to correct these technical biases, one replicate from each CI concentration experiment was set as reference, and the other replicates were linearly mapped to the same range as the reference replicate (i.e., replicate 2 as the reference).

$$O_{v,REP1\_2} = \alpha1 + \beta1 \cdot O_{v,REP1} \quad (13)$$

$$O_{v,REP3\_2} = \alpha3 + \beta3 \cdot O_{v,REP3} \quad (14)$$

In the function above, the coefficients $\alpha1$ and $\beta1$ derived from mapping the line defined by replicate 1 wild type $O_{wt,REP1}$ and weighted means of the nonsense mutations' $\hat{O}_{non,REP1}$ to the line defined by replicate 2 wild type $O_{wt,REP2}$ and weighted means of the nonsense mutations' $\hat{O}_{non,REP3}$ (Supplementary Table 6,

Fig. 15a–c).

$$\beta 1 = \frac{\hat{O}_{\text{non,REP2}} - O_{\text{wt,REP2}}}{\hat{O}_{\text{non,REP1}} - O_{\text{wt,REP1}}} \qquad (15)$$

$$\alpha 1 = \frac{\hat{O}_{\text{non,REP1}} \times O_{\text{wt,REP2}} - \hat{O}_{\text{non,REP2}} \times O_{\text{wt,REP1}}}{\hat{O}_{\text{non,REP1}} - O_{\text{wt,REP1}}} \qquad (16)$$

The same equations as (15) and (16) applies to coefficients $\alpha 3$ and $\beta 3$ to map replicate 3 to replicate 2 by only substituting replicate 1 with replicate 3.

The mean GFP signals $\hat{O}_v$ and standard errors of the mean $\widehat{OE}_v$ over biological replicates were calculated using random-effect error model for combining enrichment scores over the biological replicates.

**Calculating mutational effects at amino acid level**. In order to examine mutational effects at the amino acid level, the processed data at the nucleotide level was converted to the amino acid level.

First, for each replicate, weighted mean GFP signals of all nucleotide variants encoding the same amino acid variants were calculated. The inverse of the GFP signal errors of the nucleotide variants were given as weights. Errors from each nucleotide variants were propagated as the error of the GFP signals for each amino acid variant in each replicate.

Then, mean GFP signals $\hat{O}_v$ and the standard errors of the mean $\widehat{OE}_v$ over biological replicates at amino acid level were calculated based on the random-effect error model as for combining enrichment scores and nucleotide level GFP signals over replicates.

**Rescaling the mean GFP signals to the detection limits**. In the FACS experiments, the detection limit for the lowest GFP signal was equal to the auto-fluorescence of the bacterial cells not expressing GFP. The auto-fluorescence of the bacterial cells was not distinguishable from the cells that repressed the target gene GFP expression completely (CI WT high expression) (Supplementary Fig. 11c). The theoretical maximum GFP expression level was equal to that of bacterial cells expressing the target GFP without any repressor.

However, some variants' estimated GFP signals from the bulk sequencing data exceeded the GFP signal range defined by theoretical maximum and minimum GFP. These GFP signals outside the theoretical limits were not likely to be real and they could potentially bias our analysis.

In order to correct this problem, estimated GFP signals from the enrichment scores were rescaled to abide to the theoretical maximum and minimum GFP ranges. The lower GFP detection limit was determined by the lower limit of 95% confidence interval from the mean CI WT high expression GFP level. The upper GFP detection limit was determined by the 95th percentile of the weighted mean GFP signals of all nonsense mutations at low expression level of CI. The 95th percentile (or confidence interval) rather than the mean WT or nonsense GFP signals were selected as detection limits, so that the modes of the mutational effects would not shift after rescaling.

This GFP detection range [4.5,12.8] was first divided into 1000 evenly spaced bins ($O_k$). Then, given the observed mean GFP signal and the standard error of a variant, the probability of the true mean GFP signal of the variant falling into each bin was calculated as follows:

$$\text{pr}_{v,k} = \frac{e^{\left(-0.5 \times \left(\hat{O}_v - O_k\right)/\widehat{OE}_v\right)^2}}{\widehat{OE}_v \times 2 \times \pi^{0.5}} \qquad (17)$$

Finally, the mean GFP signal of a variant was calculated based on the weighted mean of the GFP signals from each bin with the weights given as the probability of the true mean falling into each bin $k$ ($\text{pr}_{v,k}$), as shown below:

$$\hat{O}_{v,\text{rescaled}} = \frac{\sum(\text{pr}_{v,k} \times O_k)}{\sum(\text{pr}_{v,k})} \qquad (18)$$

The $\hat{O}_{v,\text{rescaled}}$ (Supplementary Fig. 15d) was used as the mean GFP signal for each variant in the following analysis, denoted as $\widehat{O}_v$ replacing the value before transformation, and the standard error $\widehat{OE}_v$ was kept the same as before rescaling.

**Folding energy, binding energy and structural analysis**. Folding energy prediction and structural analysis were performed based on the 3.909 Å x-ray structure (PDB 3BDN) of CI dimer bound to an operator site OL1.

To estimate the mutational effects on folding energies and binding energies of CI protein, we used FoldX4 software[62]. First, BuildModel command was used to build a structural model from each single mutation in our experiment. Then, the AnalyzeComplex command (with the complexWithDNA option set to *true*) was used to obtain the absolute energies of protein-DNA complex ($\Delta G_{\text{CI-OL,FoldX}}$) as well as the protein itself ($\Delta G_{\text{F,FoldX}}$) for each mutation. Binding energy of CI to DNA ($\Delta G_{\text{B,FoldX}}$) was calculated as energy difference between the protein-DNA complex and the protein by itself for each mutation. $\Delta\Delta G$ for folding ($\Delta G_{\text{F,FoldX}}$) and binding energies ($\Delta G_{\text{B,FoldX}}$) for each variant were calculated by subtracting

folding and binding energies of wild type CI respectively.

$$\Delta\Delta G_{\text{F,foldX}} = \Delta G_{\text{F,FoldX}} - \Delta G_{\text{wt,F,FoldX}} \qquad (19)$$

$$\Delta\Delta G_{\text{B,FoldX}} = (\Delta G_{\text{CI-OL,FoldX}} - \Delta G_{\text{F,FoldX}}) - (\Delta G_{\text{wt,CI-OL,FoldX}} - \Delta G_{\text{wt,F,FoldX}}) \qquad (20)$$

Analyses were repeated with PDB structure 1LMB (1 Å x-ray structure of CI N-terminal domain bound to OL1) and with 3BDN structure bound to OR1 instead of OL1 (by mutating OL1 sequence to OR1 based on PDB 3BDN structure). FoldX4 returned the same $\Delta\Delta G$ with these analyses; therefore, only results using PDB 3BDN as a template were shown.

3D structures were visualized and analysed using PyMOL (v1.7.6.0). Amino acid positions were classified as core residues if the ratio between solvent-exposed area and the total area fell within the first quartile of the obtained data based on a PyMOL script (get_area, [https://pymolwiki.org/index.php/Get_Area]) with parameters dot_density set as 4 and dot_solvent set as 1. Positions were classified as DNA-contacting when the differences in the solvent-exposed area without DNA and with DNA were greater than 0.1 Å².

**Other features tested**. 562 amino acid indices taken from the AAindex database [https://www.genome.jp/aaindex/][63] together with BLOSUM62 matrix scores [ftp://ftp.ncbi.nih.gov/blast/matrices/], structural information, and FoldX predicted energy values were examined. The top features that correlated with the mutational effects of CI protein were: (1) the hydrophobicity index[64]; (2) the number of negative charges introduced by a mutation[65]; (3) the amino acid substitution matrix BLOSUM62[66]; (4) changes in the protein folding energy; (5) changes in the protein-DNA binding energy predicted by FoldX[62] together with the structural features of mutations (i.e., at the core, interface with DNA or at the solvent-exposed positions).

**Mathematical model**. Our aim was to build a mathematical model that captures the most important features of the system that apply to all mutations. The model propagates the effects of mutations on the folding of the lambda repressor to changes in expression of the target gene through the well-described regulatory model of the $P_R$ promoter. The model makes the following assumptions: (1) Mutations change the free energy of protein folding so altering the fraction of folded protein; (2) the fraction of folded protein is independent of the protein concentration; (3) changes in protein folding free energy are additive for all mutations. In reality, all of these assumptions may be violated for some mutations. For example, some mutations will also affect the binding affinity of the lambda repressor to the DNA operator sites or alter transcription or translation. Others may result in protein aggregation. Moreover, the fraction of folded protein may not be independent of concentration, for example at very high expression levels because of chaperone titration. Finally combining mutations in structurally contacting or indirectly energetically-coupled residues may result in non-additive changes in free energy. However, our aim was to test whether the simplest possible model of the system captured the overall changes in mutation effects and changes in the strength and sign of genetic interactions as the expression level changed. We of course acknowledge that some mutations will not meet these assumptions and these exceptions likely contribute to some of the unexplained variance in our data.

**Regulatory interaction model of the CI-repressor system**. Ackers' 8-configuration model[36] was used to predict the relationship between the total amount of CI protein and the expression levels of its repressed gene. As in our experiment, the CI regulatory interaction system in Ackers' model involves three operators (OR1, OR2 and OR3), resulting in eight possible configuration states (CS) in which the CI dimer can bind to the operators (Supplementary Table 7). Based on the model, each configuration state causes the downstream promoter to be in either an ON or OFF state (Supplementary Fig. 5a). Only two configuration states fail to repress expression of the target gene: when the CI dimer is not bound to any operators (CS1) and when CI dimer is only bound to OR3 (CS2). The probability of repressing the target gene expression is the sum of the probabilities of the six remaining configuration states that result in the OFF state of the promoter. The likelihood of each configuration state is a function of the binding energies and the free CI protein dimer concentration when the number of OR binding sites is fixed. In Ackers' model, the number of OR sites is equivalent to that found in an average lysogen (bacteria that carries the phage genes integrated in its genome) with the ORs integrated into its genome[36].

The probability that each of the eight configuration state ($f_{\text{CSi}}$) to occur is:

$$f_{\text{CSi}} = \frac{e^{-\Delta G_{\text{CSi}}/RT} \times [\text{CI}_2]^{\text{Ni}}}{\sum_i e^{-\Delta G_{\text{CSi}}/RT} \times [\text{CI}_2]^{\text{Ni}}} \qquad (21)$$

Where $\Delta G_{\text{CSi}}$ is the total free energy of lambda repressor dimers in the respective configuration $i$; the exponent $Ni$ is the total number of the lambda repressor dimers in the corresponding configuration $i$; $[\text{CI}_2]$ is the free dimer concentration; $R$ is the gas constant ($R = 1.98 \times 10^{-3}$ kcal per M) and T is the absolute temperature (310.15 kelvin).

The probability of repression ($P_s$) is the sum of the probabilities of the configurations in which promoter $P_R$ is repressed ($\sum_{i=\{3:8\}} f_{\text{CSi}}$). To calculate $P_s$ as

a function of the free dimer CI concentration $[CI_2]$ based on the equation (19) and Table S6, we obtain the following equation:

$$P_s = 1 - f_{CS1} - f_{CS2} = 1 - \frac{e^{-\Delta G_{CS1}/RT} \times [CI_2]^0 + e^{-\Delta G_{CS2}/RT} \times [CI_2]^1}{\sum_{i=1}^{8} \left( e^{-\Delta G_{CSi}/RT} \times [CI_2]^{Ni} \right)} \quad (22)$$

The target gene expression level (GFP) is modelled to be proportional to the binding probability of the RNA polymerase, which is given by one minus the probability of repression by CI ($P_s$) (Eq. (23)).

$$GFP \propto P_{RNA-pol} = 1 - P_s \quad (23)$$

Despite its simplicity, this model has been shown to be predictive of the gene expression levels[67]. Because bacterial cells displayed auto-fluorescence ($GFP_{auto}$), this auto-fluorescence signal from bacteria needed to be considered when measuring the effects of mutations on GFP levels. Therefore, by rewriting the Eq. (23) by taking into account the auto-fluorescence of the cells, the probability of GFP repression can be shown as in the Eq. (24).

$$P_s = 1 - \frac{GFP - GFP_{auto}}{GFP_{max} - GFP_{auto}} \quad (24)$$

Both Eqs. (24) and (22) show the probability of repressing the target gene, with Eq. (24) as a function of the GFP signal and Eq. (22) as a function of the free CI dimer concentration. By combining Eq. (24) with Eq. (22), we obtain an equation that describes the relationship between the free CI dimer concentration $[CI_2]$ and the GFP signal as shown in the following equation:

$$1 - P_s = \frac{GFP - GFP_{auto}}{GFP_{max} - GFP_{auto}} = \frac{e^{-\Delta G_{CS2}/RT} \times [CI_2] + 1}{\sum_{i=1}^{8} \left( e^{-\Delta G_{CSi}/RT} \times [CI_2]^{Ni} \right)} \quad (25)$$

By rewriting the Eq. (25), we can show the GFP signal as a function of free CI dimer concentration:

$$GFP = \frac{\left( (GFP_{max} - GFP_{auto}) \times \left( e^{-\Delta G_{CS2}/RT} \times [CI_2] + 1 \right) \right)}{e^{-\Delta G_{CS8}/RT} \times [CI_2]^3} + GFP_{auto}$$
$$+ \sum_{i=5}^{7} e^{-\Delta G_{CSi}/RT} \times [CI_2]^2$$
$$+ \sum_{i=2}^{4} e^{-\Delta G_{CSi}/RT} \times [CI_2] + 1 \quad (26)$$

The Eq. (26) allows us to calculate the GFP signal for each variant from a known free CI dimer concentration. In order to calculate $[CI_2]$ from Eq. (26), uniroot function was used with R script to find a unique root of Eq. (26) that was within the range of $10^{-40}$ and $10^{-3}$ M.

Next, the relationship between the total CI concentration $[CI_T]$ and the free dimer concentration $[CI_2]$ was evaluated, in order to model the relationship between the GFP signal and total CI concentration. This is because the total protein concentration in the cells but not the free dimer concentration of the protein is the one that can be experimentally measured and manipulated. The total lambda repressor concentration in the cell $[CI_T]$ is the sum of the free monomer concentration $[CI]$ plus two times the concentrations of the free dimer $[CI_2]$ plus two times the concentration of the dimers bound to operators $[OR]$. Compared to the original Ackers' model, in our experimental system, each bacterial cell was expected to carry up to hundreds of folds more operator sites, the same fold changes in CI protein coding region and the target gene. Given the same fold changes in all the functional blocks in this model, we simply kept the same parameters from original model and mapped our experimental system to the original model system.

$$[CI_T] = [CI] + 2 \times [CI_2] + 2 \times [OR_{total}] \times \sum_{i=1}^{8} (Ni \times f_{CSi}) \quad (27)$$

The concentrations of free monomer $[CI]$ and free dimer $[CI_2]$ follow the equilibrium:

$$[CI_2] \leftrightharpoons Ka \times [CI]^2 \quad (28)$$

By combining the Eqs. (27) and (28), we can describe the relationships between $[CI_T]$ and $[CI_2]$ as follows:

$$[CI_T] = ([CI_2]/K_a)^{0.5} + 2 \times [CI_2] + 2 \times [OR_{total}] \times \sum_{i=1}^{8} (Ni \times f_{CSi}) \quad (29)$$

By further substituting $\sum_{i=1}^{8} (Ni \times f_{CSi})$ from the Eq. (29) with the Eq. (21), we obtain the following equation:

$$[CI_T] = K_a^{0.5} \times [CI_2]^{0.5} + 2 \times [CI_2]$$
$$+ \frac{2 \times [OR] \times \left( \sum_{i=2}^{4} e^{-\Delta G_{CSi}/RT} + 2 \times \sum_{i=5}^{7} e^{-\Delta G_{CSi}/RT} + 3 \times e^{-\Delta G_{CS8}/RT} \right)}{\sum_{i=2}^{4} e^{-\Delta G_{CSi}/RT} \times [CI_2] + \sum_{i=5}^{7} e^{-\Delta G_{CSi}/RT} \times [CI_2]^2 + e^{-\Delta G_{CS8}/RT} \times [CI_2]^3} \quad (30)$$

The equation allows us to calculate $[CI_T]$ from $[CI_2]$. $[CI_2]$ can be calculated by finding the unique root from the Eq. (26) from the known GFP signal. Given the complexities of both Eqs. (26) and (30), the calculations were performed in two steps according to the two equations. For the following process, for ease of

reference, we denote the process of calculating total protein $[CI_T]$ for each variant from its target GFP signals $f'_{Ackers}$:

$$[CI_T] = f'_{Ackers}(GFP) \quad (31)$$

The reverse process to calculate GFP signals from the total protein $[CI_T]$ involves two steps: (1) inversing Eq. (30) to calculate the corresponding $[CI_2]$; (2) calculating GFP with Eq. (26) from the previous step.

Inversing and finding the exact root of Eq. (30) is mathematically impossible. Therefore, an approximate solution was found based on a local polynomial regression (loess function with R, span parameter 0.3) describing the relationship between $[CI_2]$ and $[CI_T]$ based on Eq. (30) (Supplementary Fig. 16a).

Based on Eq. (26), GFP signal was calculated by inputting $[CI_2]$ from the previous step. We denote the process as $f_{Ackers}$, which is the inverse of Eq. (31) for the ease of future reference.

$$GFP = f_{Ackers}([CI_T]) \quad (32)$$

The parameters were kept as they were originally used in the model by Ackers[36] (Supplementary Table 8) that were experimentally determined.

Two additional parameters ($GFP_{max}$ and $GFP_{auto}$) were specific to our experiment and not described in the original model by Ackers. For modelling, the maximum GFP signal $GFP_{max}$ was defined as the weighted mean GFP signals of all single nonsense mutations with weights given as the inverse of the variance (3470.67 AU, or 11.76 AU in log2 scale) based on the CI low expression dataset. The minimum GFP signal $GFP_{auto}$, corresponding to the cellular auto-fluorescence GFP signal, was found through parameter search as follows. To start with, two constraints for $GFP_{auto}$ were considered: first, based on the regulatory interaction model, repression of the target gene expression can never reach 100% even though it can infinitely approach this level.

In other words, the $GFP_{auto}$ cannot be set to be the same as the GFP signal from the wild type protein at high expression. Second, $GFP_{auto}$ should allow the calculated ratio of wild type $[CI_T]$ between high and low expression levels based on Eq. (29) $f'_{Ackers}$ to agree with experimentally quantified ratio (15:1, see protein quantification section, Supplementary Fig. 12). We performed the parameter search for $GFP_{auto}$ that allowed the ratio of calculated wild type $[CI_T]$ at two expression levels to be 15:1 based on the model calculation as shown below:

$$\frac{\left[ CI_{T,wt,High} \right]}{\left[ CI_{T,wt,Low} \right]} = \frac{f'_{Ackers} \left( GFP_{wt,High} \right)}{f'_{Ackers} \left( GFP_{wt,Low} \right)} = 15 \quad (33)$$

$GFP_{auto}$ was estimated to be 23.24 AU (4.54 AU in log2 scale) to meet the condition set by Eq. (33).

**Estimating the functional protein concentration**. An estimate of wild type CI protein concentration $[CI_{T,wt}]$ in each of the two experiments can be obtained by inputting $GFP_{wt,High}$ and $GFP_{wt,Low}$ values into $f'_{Ackers}$ function. The same way, the total protein concentration of a variant $[CI_{T,v}]$ can be derived for each experiment with the $f'_{Ackers}$ function. Differences between $[CI_{T,v}]$ and $[CI_{T,wt}]$ were assigned to differences in their functional protein fraction rather than changes in the total expressed protein amount. This is based on the assumption that the mutations with one or two amino acid alterations affected GFP levels mostly through changing the fractions of natively folded protein ($f_N$).

In order to calculate the fraction of correctly folded protein for each variant ($f_{N,v}$), knowledge of the total expressed protein concentration $[CI_E]$ for each experiment was needed. Based on the calculated $[CI_{T,wt}]$ at both low and high expression levels and the information that the fraction folded of the wild type protein is 0.9913 (see the next section), the total expressed protein concentration in the cell can be calculated by dividing the concentration of functional wild type CI protein by 0.9913 (Supplementary Table 9).

The fraction of natively folded protein for a variant $v$ ($f_{N,v}$) was calculated as the ratio of $[CI_{T,v}]$ (that is calculated based on $f'_{Ackers}$ ($GFP_v$)) over total expressed CI $[CI_E]$ (as a parameter calculated based on $f'_{Ackers}$ ($GFP_{wt}$), Table S8):

$$f_{N,v} = \frac{\left[ CI_{T,v} \right]}{[CI_E]} \quad (34)$$

**Thermodynamics of CI folding model**. CI has been shown to follow a two-state model of protein folding[68] that can be described with the following equation:

$$f_N = f_{folding}(\Delta G_F) = \frac{e^{-\frac{\Delta G_F}{R \cdot T}}}{1 + e^{-\frac{\Delta G_F}{R \cdot T}}} \quad (35)$$

With $f_N$ as the fraction of natively folded protein, $\Delta G_F$ as the total free energy of the protein folding. $R$ is the gas constant ($R = 1.98 \times 10^{-3}$ kcal per M) and $T$ is the absolute temperature of our experimental setting ($T = 310.15$ kelvin, 37 °C). Rewriting the Eq. (35), we obtain:

$$\Delta G_F = f'_{folding}(f_N) = -R \times T \times \ln\left( \frac{f_N}{1 - f_N} \right) \quad (36)$$

The equilibrium between the concentration of unfolded and native CI protein follows the equation below:

$$\text{CI}_U \rightleftharpoons \text{CI}_N \tag{37}$$

Eq. (37) is governed by an equilibrium constant $K_{fold}$ whose value is known to be 114 for the wild type CI protein[69]:

$$K_{fold} = \frac{[\text{CI}_N]}{[\text{CI}_U]} = \frac{f_N}{1 - f_N} = e^{-\frac{\Delta G_F}{RT}} \tag{38}$$

By solving Eq. (38) with $K_{fold} = 114$, we obtain the wild type CI $f_N = 0.9913$ which was used to calculate the total protein concentration in the cells (Supplementary Table 9), as shown in the previous section.

The folding energy of a double missense mutation (AB) can be predicted by adding the folding energies of the two single mutations (A and B) that together make the double mutation (AB).

$$\Delta G_{F,AB,predict} = \Delta G_{F,A} + \Delta G_{F,B} - \Delta G_{F,wt} \tag{39}$$

**Combining models**. To predict GFP signal of a mutation $A$ from $\Delta G_F$ values of the mutation, the output of $f_{folding}$ function was added to $f_{Ackers}$ function:

$$\text{GFP}_A = f_{Ackers}\left(f_{folding}\left(\Delta G_{F,A}\right) \times [\text{CI}_E]\right) \tag{40}$$

$$\Delta G_{F,A} = f'_{folding}\left(\frac{f'_{Ackers}(\text{GFP}_A)}{[\text{CI}_E]}\right) \tag{41}$$

**Comparing four different sub-models**. To evaluate the importance of (a) protein folding and (b) CI-concentration-dependent repression of the target gene expression independently as well as in combination, we generated and compared four models (Fig. 3, Supplementary Fig. 5). The four models are based on four different assumptions. The first model is the log-additive model where changes in the target gene expression levels are simply additive in the log scale (Supplementary Fig. 5c). The second model is the full model that incorporates the effects of mutations both at the level of protein folding and at the level of regulatory interaction of CI-OR system on the target gene expression (as shown in Eqs. (39–41), Fig. 3a, Supplementary Fig. 5b, c). The third model is a protein folding-only model that incorporates the thermodynamics of protein folding but not the regulatory interaction model (it assumes a linear relationship between target gene expression and functional CI concentration). Therefore, the protein folding energies are additive features of this model (Supplementary Fig. 5b–d). The last model is the regulation-only model that incorporates the regulatory interaction model but not the thermodynamics of protein folding (it assumes a linear relationship $\Delta G_F$ and $f_N$). Therefore, the functional protein amount is the additive feature of this model (Supplementary Fig. 5b, c, e).

Depending on the model evaluated, the functions linking the target gene GFP expression level to $[\text{CI}_T]$, or $[\text{CI}_T]$ to $\Delta G_F$ can be different. The details of each model are explained below.

**Log-additive model**. Consistent with extensively used null models where the effects of mutations are log-additive, this model predicts the log GFP signal of a double mutation $AB$ relative to the wild type to be the sum of the log GFP signals of each of the two single mutations relative to the wild type:

$$\log_2\left(\text{GFP}_{AB,predicted}\right) - \log_2(\text{GFP}_{wt})$$
$$= \left(\log_2(\text{GFP}_A) - \log_2(\text{GFP}_{wt})\right) + \left(\log_2(\text{GFP}_B) - \log_2(\text{GFP}_{wt})\right) \tag{42}$$

Therefore,

$$\log_2\left(\text{GFP}_{AB,predicted}\right) = \log_2(\text{GFP}_A) + \log_2(\text{GFP}_B) - \log_2(\text{GFP}_{wt}) \tag{43}$$

**Full model**. To predict the GFP expression levels of a double mutation, we first estimated the $\Delta G_F$ of the corresponding single mutations using Eq. (41). The $\Delta G_{F,AB}$ of the double mutation was calculated with Eq. (39) and then converted to an expected GFP signal using Eq. (40).

**Folding-only model**. This model assumes that the GFP expression levels are linearly responsive to the fraction of natively folded protein $f_N$. That is, this model replaces $f_{Ackers}$ with a linear transformation between GFP signal and $f_N$ (Supplementary Fig. 5e). At the same time, this model includes the nonlinear relationship between $f_N$ and $\Delta G_F$ that was introduced by the thermodynamics model of protein folding. Thus, for a mutation $A$, the relationship between GFP signal and the fraction of folded protein $f_N$ was given by a modified version of $f_{Ackers}$, which we

call $f_{model3}$.

$$\log_2(\text{GFP}_A) = f_{model3}\left(\log_2\left([\text{CI}_{T,A}]\right)\right) = \alpha + \beta \times \log_2\left([\text{CI}_{T,A}]\right) \tag{44}$$

$$\log_2\left([\text{CI}_{T,A}]\right) = f'_{model3}\left(\log_2(\text{GFP}_A)\right) = \frac{\log_2(\text{GFP}_A) - \alpha}{\beta} \tag{45}$$

The output of $f_{model3}$ can then be introduced into $f'_{folding}$ (Eqs. 36 and 39).

$$\Delta G_{F,A} = f'_{folding}\left(\frac{f'_{model3}(\text{GFP}_A)}{[\text{CI}_E]}\right) \tag{46}$$

$$\text{GFP}_A = f_{model3}\left(f_{folding}\left(\Delta G_{F,A}\right) \times [\text{CI}_E]\right) \tag{47}$$

The $\alpha$ and $\beta$ parameters from $f_{model3}$ (Eq. (44)) determine the linear relationship between the functional repressor concentration and GFP expression levels ($\alpha$ is the intercept and $\beta$ is the slope). Also, the parameters $[\text{CI}_{E,low}]$ and $[\text{CI}_{E,high}]$ (Supplementary Table 9) were kept the same as in the other models.

Comparing the mutational effects at two expression levels based on Eq. (44), we obtain the following equation:

$$\log_2\left([\text{GFP}_{A,high}]\right) - \log_2\left([\text{GFP}_{A,low}]\right) = \beta \times \log_2\left(\frac{[\text{CI}_{T,A,high}]}{[\text{CI}_{T,A,low}]}\right) \tag{48}$$

The ratio of $[\text{CI}_{T,A}]$ at two expression levels was set as the constant 15 (as defined by wild type protein, see the previous section). Eq. (48) therefore can be re-written as follows:

$$\log_2\left(\text{GFP}_{A,high}\right) = \beta \times \log_2(15) + \log_2\left(\text{GFP}_{A,low}\right) \tag{49}$$

By substituting $\beta \times \log_2(15)$ with a coefficient $C$, we can rewrite Eq. (49) as follows:

$$\log_2\left(\text{GFP}_{A,high}\right) = C + \log_2\left(\text{GFP}_{A,low}\right) \tag{50}$$

From Eq. (50), we can see that GFP signal at the two CI expression levels is linearly related with the fixed slope of one in the log space. Parameter search was performed to find the coefficient $C$ that best described the observed relationships between GFP signals at low and high expression levels of CI. In detail, we firstly sampled a hundred $\log_2(\text{GFP}_{v,low})$ values ranging between $\log_2(\text{GFP}_{wt,low}) = 7.23$ and $\log_2(\text{GFP}_{max}) = 11.76$. Then, a range of intercept $C$ between $-3.3$ and $-1.3$ with the step of 0.03 was used to calculate the corresponding $\log_2(\text{GFP}_{high})$ for each $\log_2(\text{GFP}_{low})$ (Supplementary Figure 16b - d). The value $C = -2.07$ was selected that resulted in the smallest sum of the squared distances from the observed data points to the line defined by simulated relationship between GFP signals at two expression levels (Supplementary Fig. 16c). Based on $C = \beta \times \log_2(15)$, we further calculated $\beta = -0.52$.

The coefficient $\alpha$ was calculated by placing the values $\beta$ and the wild type CI low expression data $[\text{CI}_{E,low}]$ and $\log_2(\text{GFP}_{wt,low})$ to Eq. (44) and rewritten as below:

$$\alpha = \log_2\left(\text{GFP}_{wt,low}\right) - \beta \times \log_2\left(f_{N,WT} \times [\text{CI}_{E,low}]\right) \tag{51}$$

With $\log_2(\text{GFP}_{wt,low}) = 7.23$ as observed in the experiment, $\beta = -0.52$ as calculated above, and the known parameters $f_{N,wt} = 0.9913$ and $[\text{CI}_{E,low}] = 5.5 \times 10^{-8}$, we obtained $\alpha = -17$.

To estimate the GFP signals of a double mutations with the folding-only model, we first estimated the $\Delta G_{F,A}$ and $\Delta G_{F,B}$ of the corresponding single mutations using Eq. (46). Double mutants' $\Delta G_{F,AB}$ was calculated using Eq. (39). $\Delta G_{F,AB}$ of the double mutant was then converted to an expected GFP signal using Eq. (47) (Supplementary Fig. 5b, c).

**Regulation-only model**. This model assumes that the $f_N$ of a protein is linearly related to its $\Delta G_F$. That is, this model replaces $f_{folding}$ with a linear transformation between $f_N$ and $\Delta G_F$. At the same time, this model includes the nonlinear relationship between $f_N$ and GFP expression levels from Ackers' model.

Because of the assumed linear relationship between $f_N$ and $\Delta G_F$, the effects of mutations are additive in $f_N$ space making $f_{folding}$ unnecessary in this model at all (Supplementary Fig. 5b,c). To estimate the GFP expression levels of a double mutant with the regulation-only model, we first estimated the functional protein concentration of the corresponding single mutants using $f'_{Ackers}$ (GFP). The expected functional protein concentration of the double mutant was then given by the following equation.

$$[\text{CI}_{T,AB}] = [\text{CI}_{T,A}] + [\text{CI}_{T,B}] - [\text{CI}_{T,wt}] \tag{52}$$

The expected GFP signal for this double mutant was calculated using $f_{Ackers}$ ($[\text{CI}_{T,AB}]$), as shown in Eq. (32) (Supplementary Fig. 5b, c).

**Simulation overview**. To test to which extent each model can explain (1) the double mutational effects given the single mutational effects (2) the relationship between the mutational effects at the two protein concentrations (3) the pair-wise

genetic interactions at both protein concentrations, we simulated mutational effects and their interactions based on each model to compare with our data.

**Simulating the mutational effects based on the model**. We sampled 100 $\Delta G_F$ values equally spaced between $-3$ kcal per mol and 3 kcal per mol, and estimated their GFP signals at high and low CI concentrations using each of the four sub-models described above. For a given model, plotting the GFP signals predicted for the high CI concentration case against the GFP signals predicted for the low CI concentration case resulted in a curve (or a line) (Fig. 3e).

To test how well each model explained the observed protein concentration-dependent mutational effects, we used the Princurve package[70] in R to calculate the sum of squared distance from the curve (SSDC) between every experimental data point and the line or curve described by the model (Supplementary Fig. 16e).

To predict the $\Delta G_F$ for a specific variant, we first projected each data point in the log GFP (high CI concentration) vs. log GFP (low CI concentration) scatter plot to the nearest point in the model curve (line, in the case of folding-only model) (Supplementary Fig. 16d, e). The projected GFP signal corresponds to a single $\Delta G_F$ of each variant based on the model. This correction allowed us to estimate a single $\Delta G_F$ using the GFP value for both the high and the low CI concentrations. Finally, this estimated $\Delta G_F$ (functional protein concentration [$CI_T$], in the case of regulation-only model) of a variant was used in the following processes for predicting double mutational effects and to predict the epistasis patterns.

**Comparing model predicted and observed double mutational effects**. The percentage of variance explained (PVE) for the mean GFP expression levels of the double mutation was calculated as follows:

$$PVE = \left(1 - \frac{SS_{res}}{SS_{Total}}\right) \times 100 \tag{53}$$

Where $SS_{res}$ is the residual sum of squares between the model-predicted versus the observed GFP expression levels and $SS_{Total}$ is the variance in the observed data.

**Predicting pair-wise genetic interactions with each sub-model**. Epistasis was defined as the difference between the GFP expression levels of a double mutant based on the model (full model, folding-only model and regulation-only model) and the log-additive model (Eq. (43)), as shown in the equation below (Fig. 2f):

$$Epistasis_{Model\_i} = \log_2\left(GFP_{AB,log-additive}\right) - \log_2\left(GFP_{AB,Model\_i}\right) \tag{54}$$

For a given double mutant, we first predicted the $\Delta G_F$ values (full model and the folding-only model) or $f_N$ (regulation-only model) of the corresponding single mutants, as stated above. We then used each model to convert the double mutant's predicted $\Delta G_F$ or $f_N$ value back into the GFP signal. This predicted GFP signal was compared with the expected GFP signal based on the log-additive null model (Eq. (43)). The genetic interaction patterns were further compared to the experimental observation (Fig. 3g–k and Supplementary Figs. 6, 7).

The summary of the modelling mutational effects based on each model was illustrated as a cartoon in the Supplementary Fig. 5a–c.

**Toy models of three protein expression–fitness relationships**. Three most common fitness-protein concentration relationships were modelled based on the fitness effects of changes in protein concentrations in yeast[49].

Fitness increases with lower protein concentration:

$$\omega_I = \frac{[protein]}{(0.1 + [protein])} \tag{55}$$

Fitness with optimal protein concentration:

$$\omega_O = \frac{1.2 \times [protein]}{(0.1 + [protein])} \times \frac{1}{(1 + 0.1 \times [protein])} \tag{56}$$

Fitness decreases with higher protein concentration:

$$\omega_D = \frac{1}{1 + 0.1 \times [protein]} \tag{57}$$

These functions were integrated into the full model (sub-model 2) in place of $f_{Ackers}$ to build three new models linking fitness to changes in protein folding energy $\Delta G_F$. Note that because $f_{folding}$ was left untouched, all these three new models assumed a two-state protein folding kinetics as for CI protein. Mutational effects and pairwise genetic interactions were analysed at two simulated protein concentrations (high and low) based on these models. For both 'Increasing' and 'Decreasing' fitness landscapes, the two wild type protein concentrations in each simulation were selected so that one wild type protein concentration would be abundant enough to be robust to mutational effects and the other one would be sensitive to the mutational effects. For the 'Peaked fitness landscape', the two protein concentrations were selected so that the fitness effects would be the same but the protein expression levels at 'Low' would be below the optimal protein concentration and at 'High' would be above the optimal protein concentration.

We evaluated the effects of 50 mutations with $\Delta\Delta G_F$ evenly spaced between $-1$ kcal per mol and $+5$ kcal per mol in four different wild type proteins with different protein folding energies: (1) very stable wild type protein ($\Delta G_{F,wt} = -3$ kcal per mol); (2) stable wild type protein ($\Delta G_{F,wt} = -1.6$ kcal per mol); (3) marginally stable wild type protein ($\Delta G_{F,wt} = -1$ kcal per mol); (4) unstable wild type protein ($\Delta G_{F,wt} = 0$ kcal per mol) (Fig. 5a–h, S7). The effects on fitness of all the pairwise combinations of mutations were also evaluated assuming that the effects of mutations are additive in $\Delta\Delta G_F$ space.

Epistasis was quantified as the difference between the observed double mutational effects (calculated by adding the $\Delta\Delta G_F$ of the single mutations,) with the expected effects (calculated by adding up single mutational effects based on the log-additive null model):

$$Epistasis = Observed_{Fitness} - Expected_{Fitness} \tag{58}$$

**Quantification and statistical analysis**. Statistical details of experiments including the statistical test used, the exact number of the data points, mean values, standard errors of the mean (s.e.m.), and 95% confidence intervals, p-values can be found in the figure legends and results. Data with low reproducibility from the three biological replicates (s.e.m > 1 for the predicted mean GFP) were excluded from subsequent analyses.

**Reporting summary**. Further information on research design is available in the Nature Research Reporting Summary linked to this article.

## Data availability

Processed data used for the analysis is available as Supplementary Data 1. Raw Illumina sequencing data and the processed count data files that support the findings of this study have been deposited in NCBI's Gene Expression Omnibus and are accessible through GEO Series accession number GSE122806 [https://www.ncbi.nlm.nih.gov/geo/query/acc.cgi?acc=GSE122806].

## Code availability

Scripts are available from GitHub [https://github.com/lehner-lab/concentration_epistasis_CI].

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

## Acknowledgements

We thank members of the Lehner lab and J. Ren for comments on the paper. This work was supported by a European Research Council (ERC) Consolidator grant (616434), the Spanish Ministry of Economy and Competitiveness (BFU2017-89488-P and SEV-2012-0208), the Bettencourt Schueller Foundation, Agencia de Gestio d'Ajuts Universitaris i de Recerca (AGAUR, 2017 SGR 1322.), and the CERCA Program/Generalitat de Catalunya. X. Li was supported in part by a fellowship from the Ramón Areces Foundation. We also acknowledge the support of the Spanish Ministry of Economy, Industry and Competi-tiveness (MEIC) to the EMBL partnership and the Centro de Excelencia Severo Ochoa.

## Author contributions

X.L. performed all experiments, analyses and modelling. J.L. built the plasmid construct pCIPR. X.L., J.L., R.D. and B.L. conceived the project. X.L. and B.L. designed the project and interpreted the data, X.L., P.B-C. and B.L. wrote the paper.

## Additional information

**Competing interests:** The authors declare no competing interests.

