## [Peer Review File · Nature Communications]

Reviewers' Comments:

Reviewer #1:

Remarks to the Author:

The authors study the epistasis behaviour of a minimal gene network involving a single repressor protein inhibiting a target gene, where they deliberately vary the expression level of the repressor and introduce mutations in the repressor protein affecting the binding properties of the repressor to the target gene promoter. There is an enormous literature on epistatic gene networks, but I think this is the first time anyone experimentally reveals the complexity of the genotype-phenotype map when one combine variation in the concentration of a regulatory protein with variation of its binding properties. Even though the experimental setup is very simple it is still a system accounting for up-stream gene regulatory dynamics affecting the expression level of the repressor protein, the molecular dynamics underlying the folding dynamics and functional properties of the repressor as function of mutations in its coding region, and the promoter binding dynamics to the target gene promoter. Thus, it is a highly representative regulatory situation.

The production of the experimental results and the linking to a mathematical model capable predicting the combined effect of variation in expression and functional properties of the repressor protein, has demanded quite an effort. Thus there are limits to how much more one should demand from a paper like this. However, I think the paper as it is misses an opportunity window by not making a clearer link to the theoretical literature on epistatic gene networks and the propagation and penetrance of mutational effects in gene networks as a function of mutational background. In fact, if the paper had been pitched as an experimental test of a set of theoretical predictions about how their experimental system would behave, it would have made an even stronger impression on me even if Sir Peter Medawar's claim that "the scientific paper is a fake" would apply.

The space of nonlinear functions is vast, and for the sake of clarity it would be good if the authors qualified their use of the term "nonlinear relationship" upfront as long as they deal with a tiny subsets of nonlinear functions, in fact the three that are commonly dealt with in connection with gene regulatory networks. The paper would not miss any generic value by doing this. There is a rich gene network dynamics literature describing the effects of steep sigmoidal and non-monotonic dose-response relationships, and it would be good if the authors place their results in the context of this literature a little more.

In the experimental system under study the combined effect of two mutations is realised by molecular dynamics at two levels, as there is a protein phenotype emerging from intragenic mutations. One therefore has an additional GP map in the system which the mathematical model makes use of. The existence of an additional lower-level GP map enlarges the empirical scope of the paper, and it could be worthwhile to reflect on the implications of this in connection with multiscale biology.

What I find most intriguing with the paper is that the system under study shows such complex sign epistasis behaviour. This is a phenomenon that is of considerable significance in biomedicine, production biology and evolutionary biology. And I have to admit that I was a bit surprised that even this simple system can display so much sign epistasis. Sign epistasis is caused by order breaking of the GP map, i.e. the partial order of genotypes breaks with the order of the associated phenotypes such that GP map becomes non-monotonic. Monotonicity is a mathematical term wherefrom we can develop theory while sign epistasis is just a descriptive term caused by the underlying non-monotonicity. The authors do not qualify this link between monotonicity and sign epistasis. Thus I think it would be a worthwhile exercise to interpret the results in terms of monotonic and non-monotonic features of the GP map revealed by the experimental study. It has been shown, by using a monotonicity measure for many-loci systems, that in a wide class of gene regulatory networks most GP maps are monotonic but that the presence of positive feedback and incoherent feedforward predisposes a network to possess non-monotonic GP map features. As the experimental system under study does neither contain positive feedback loops nor incoherent

feedforward motifs, the results certainly bring new insight to the table in terms of how frequent non-monotonic GP maps might arise. I think it would strengthen paper if the authors made this point clear.

Overall, I think this paper is important for providing new data on the complexity of epistasis patterns that exist even in a very simplified setting and which transfers to many domains, it shows how a quite simple mechanistic model can explain much of this complexity, it demonstrates a new mechanism for creating non-monotonic GP maps, and it provides a theoretical-experimental foundation for systematic elucidation of how the GP map changes as a function of mutational changes affecting both expression and structure of regulatory proteins. For these reasons, the results should be of interest to the community and the wider field.

I am confident that the results are reproducible as the description of the experimental and theoretical work is pretty well articulated.

Minor comments:

Line 24: There are many aspects connected to the challenge of making genetic predictions. Consider to change the phrasing "To understand the reasons for this" to something more moderate and informative, *we wish to reveal how intricate the GP map might be even in a very simple gene regulatory context.*

Line 26: The term "genetic interaction" is loaded with different interpretational content depending on which field you come from. The term comes from statistical genetics but is used widely as shorthand in other communities, but with different interpretations. However, the mutations as such do not interact in the sense that they have a reciprocal effect on each other as such. Their interdependence is manifested in the GP map, and we would benefit from using more precise wording as long as we want to develop causal explanatory structures.

Line 59: Skip Why is this? It is redundant.

Some headings in the Results section make generic claims that are not original. Consider some more specific wording.

Skip the first two paragraphs in Discussion. Instead of telling me what you have just told me, use the space to qualify possible shortcomings and outline the generic value of the results. Start with the possible shortcomings to meet the readers questions up front.

Line 267: The plasticity of epistasis is not an original finding. There is a rich literature on this so either you contextualise or you skip the paragraph. I suggest the latter.

Line 272: There already exist theoretical literature on the propagation of mutational effects across scales and in gene regulatory networks. Place your claim and results in this paragraph in the existing context or skip the paragraph.

Best regards,

Stig W. Omholt

PS. Please contact me for references if needed.

Reviewer #2:

Remarks to the Author:

Thank you for the opportunity to review "Changes in gene expression predictably shift and switch genetic interactions" by Li et al. This manuscript describes how varying expression level of a gene changes the direction and magnitude of single mutant effects as well as pairwise genetic interactions between mutations. Authors use phage lambda repressor (CI), a well-studied protein and a model for gene regulation, expressed at two different levels (low and high) to test the effect of single and pairwise mutations on its transcriptional activity. They use 'doped' oligonucleotide synthesis to randomly introduce mutations into the 59 amino acid helix-turn-held DNA-binding domain of CI and use GFP fluorescence intensity as a readout for the transcriptional activity of CI variants. They use FACS to separate cells into two bins with neutral and intermediate population based on GFP fluorescence and then use deep sequencing to identify the variants in each bin. Authors then quantify single mutant effects as well as genetic interactions between mutations and generate a model for their prediction by integrating protein folding and protein concentration. They show that changes in gene expression can affect the magnitude and the direction of genetic interactions between mutations.

The idea to investigate the effect of how gene expression could modify the effects of single and pairwise combinations of mutations within a gene of interest is interesting and important. Other studies examine the effect of mutations singly and in combination on protein function by studying a gene of interest at a defined expression level, whereas the novelty of this study is the comparison of these effects between two different gene expression levels. This is an elegant study and the manuscript is well-written and logical. I only have minor suggestions that the authors should consider in a revised manuscript.

1. On page 4 authors should explain how 18 single and 4 double mutants were chosen to compare the behaviour of mutants between bulk sequencing and individual testing.
2. On the same page, authors claim that mutations to less similar amino acids were more detrimental and mutations that introduced a negative charge were also more detrimental at DNA contact sites, but it is unclear if this difference is statistically significant since Figure S2C, S2D, S2I and S2J do not show results of any statistical tests.
3. The data presented in Figure S2E and S2F are not convincing since most points are oriented vertically suggesting that delta G folding or delta G binding shows no correlation with GFP intensity and that the correlation is driven by 5 outliers (in case of delta G binding). It may be helpful to separate the plots into solvent-exposed, core and contacting DNA sub-plots otherwise because the points overlap, it is difficult to see a trend. It may also be helpful to bin delta G and use additional complementary methods to calculate delta G folding and binding to make robust conclusions.
4. The Methods section on 'Sorting' on pg 19 describes 3 gates (near neutral, completely detrimental and intermediate) that were used to sort cell populations. Authors should explain why they only used near neutral and intermediate populations for deep sequencing.
5. In Figure 2, the total number of interactions in 4 different classes does not add up to 100% but to 93%. Authors should account for the difference. It is surprising that there are no points in the top left quadrant of Figure 2D and 2E? There are no CI variants that have no effect at low expression but are detrimental at high expression? Would this subset be observed if higher order interactions were examined or is this a technical limitation of the experiment or the quantification?
6. The model, which is described in Figure 3, includes protein folding and protein concentration as parameters to predict the effect of mutations. However, it is known that CI easily dimerizes, so it would be prudent to include the effect of mutations on dimerization status/kinetics into the model.
7. It would be helpful to discuss higher order genetic interactions and to what extent they are going to be modified by gene expression changes. Additionally, the majority (62%) of interactions does not change due to differences in CI gene expression. Authors should not lose the sight of the stability of genetic interaction networks.
8. Supplementary figures should be called out sequentially so on pg 5 Figure S5A is referenced immediately after Figure S2. It may be prudent to move it up and rename it to Figure S3 to enable easy navigation through the manuscript.

Reviewer #3:

Remarks to the Author:

Summary:

The authors use the phage lambda repressor (CI) as a model to study how the effects of mutations, and their interactions, change across two different expression levels. Having constructed a mutant plasmid library, where every plasmid contains GFP and a mutagenized CI, they use fluorescence as a readout of CI ability to repress GFP. After fluorescence-activated cell sorting into two bins, the mutagenized region is sequenced, and the enrichment of reads in every bin is used to predict the GFP signal of every mutant CI genotype. They then go on to estimate single mutation and interaction effects, and show how the differences in these effects across the two expression levels is predicted by a mathematical model involving folding and binding energies of CI.

Overall evaluation:

I find this manuscript very interesting. The point about genetic interactions emerging as a consequence of a non-linear relationship between traits, such as gene expression and fitness, has been made before. However, the fact that these effects will also depend on the expression “baseline” has to my knowledge not been articulated previously. The authors also use an appealing model system (the phage lambda repressor) to empirically study this phenomenon which is a novel contribution, since this has previously largely been a theoretical argument. Finally, the point about non-monotonic expression-phenotype relationships resulting in ambiguous phenotypic predictions raises many interesting questions. Overall, I think this study makes novel contributions that might be of general relevance for our understanding of genotype-phenotype relationships, and ultimately our ability to predict phenotypes from genotypes. However, I’m not fully convinced that the empirical results presented support the main claims (see detailed comments).

Major Comments:

1) My first major concern regards the writing of the manuscript. I found it impossible to understand how the experiment was done based on the results section alone. Many crucial details have been left out, such as the fact that the GFP expression per genotype that the authors use in subsequent analyzes are not directly observed, but indirectly inferred from enrichment of reads in the two sorted bins. The reader has to dig quite deep in the long methods section in order to understand what the authors did, which makes this a very “heavy” read. I would appreciate an extended version of fig1A-C describing all the major steps in the experiment, and a few more details in the results section.

2) The analysis in this manuscript involves a rather long chain, where an indirectly inferred metric is used to infer another metric and so on. My biggest concern here is the GFP level per mutant genotype, which most subsequent analyzes depend upon. This is inferred from the amount of enriched reads in the two sorted bins. In order to validate this method, the authors individually quantify the GFP levels of 22 genotypes and compare it to the predicted GFP (Fig1D). Although the prediction looks good in these cases, the 22 genotypes were picked because they were highly reproducible in terms of read enrichment. To me, this does not sound like a fair representation of the mutant library. Would it be possible to perform the same validation with a random sample of mutant genotypes?

3) The authors compare four different mathematical models and conclude that the “full model”, which allows for non-linearity both in protein folding and regulatory interaction, best fits the data. While this does sound reasonable, the full model should also be most flexible to fit any data since it has more parameters. I’m thus suspicious that the better fit is at least in part due to overfitting. Would it be possible to perform some sort of out of sample prediction, where test and training data are kept separate, to validate these models?

Minor Comments:

1) The authors point out that double mutations with high expected target gene expression tended

to interact positively and vice versa (lines 139-143). Isn't this expected per definition? They define positive interactions as less target gene expression than expected (Fig 2F). So this should by definition happen when the expectation is high right?

2) Why did the authors not sequence genotypes in the "detrimental gate" after sorting? Economic reasons? Wouldn't an enrichment score based on all three bins, rather than just two, give a better prediction of GFP levels?

Response to the reviewers' comments

We thank all the reviewers for their insightful comments and constructive suggestions. Please see below for our detailed responses to each suggestion.

Reviewer #1 (Remarks to the Author)

The authors study the epistasis behaviour of a minimal gene network involving a single repressor protein inhibiting a target gene, where they deliberately vary the expression level of the repressor and introduce mutations in the repressor protein affecting the binding properties of the repressor to the target gene promoter. There is an enormous literature on epistatic gene networks, but I think this is the first time anyone experimentally reveals the complexity of the genotype-phenotype map when one combine variation in the concentration of a regulatory protein with variation of its binding properties. Even though the experimental setup is very simple it is still a system accounting for up-stream gene regulatory dynamics affecting the expression level of the repressor protein, the molecular dynamics underlying the folding dynamics and functional properties of the repressor as function of mutations in its coding region, and the promoter binding dynamics to the target gene promoter. Thus, it is a highly representative regulatory situation.

We thank reviewer #1 for the enthusiasm and insightful and helpful comments. Please see below for our replies to the suggestions and comments.

The production of the experimental results and the linking to a mathematical model capable predicting the combined effect of variation in expression and functional properties of the repressor protein, has demanded quite an effort. Thus there are limits to how much more one should demand from a paper like this. However, I think the paper as it is misses an opportunity window by not making a clearer link to the theoretical literature on epistatic gene networks and the propagation and penetrance of mutational effects in gene networks as a function of mutational background. In fact, if the paper had been pitched as an experimental test of a set of theoretical predictions about how their experimental system would behave, it would have made an even stronger impression on me even if Sir Peter Medawar's claim that "the scientific paper is a fake" would apply.

We thank the referee for this suggestion and have included a more substantial introduction to the theoretical literature in the introduction of the revised manuscript. However, we have kept the current structure of the results because it does reflect the reality of how this project proceeded – we performed the experiment, found the data very puzzling and were only able to make sense of it after we combined the existing thermodynamic model of regulation by the lambda repressor with the thermodynamic model of protein folding.

We have added the following text to the introduction:

"There is a rich theoretical literature on how both biochemistry and regulatory networks can generate many of the classic 'statistical' phenomena of

genetics³⁰⁻³³, including interactions between mutations^{32,34,35}. For example, the thermodynamics of protein folding³⁴ and molecular interactions² result in non-linear relationships between changes in free energy and the activity of individual molecules and complexes. Similarly regulatory systems often have steep sigmoidal dose-response functions because of cooperativity, molecular titration and feedback^{33,36}. The kinetic coupling of enzymes can also generate non-linear expression-fitness functions³⁷. Thus, pioneering theoretical work has shown that many mechanistic aspects of molecular biology are expected to produce non-additive genetic interactions between mutations³⁵. ”

The space of nonlinear functions is vast, and for the sake of clarity it would be good if the authors qualified their use of the term “nonlinear relationship” upfront as long as they deal with a tiny subsets of nonlinear functions, in fact the three that are commonly dealt with in connection with gene regulatory networks. The paper would not miss any generic value by doing this. There is a rich gene network dynamics literature describing the effects of steep sigmoidal and non-monotonic dose-response relationships, and it would be good if the authors place their results in the context of this literature a little more.

As stated above, we have now introduced more of the inspiring theoretical literature in the introduction. Throughout the text we have also been more specific about the (shape of) nonlinear functions we are describing.

In the experimental system under study the combined effect of two mutations is realised by molecular dynamics at two levels, as there is a protein phenotype emerging from intragenic mutations. One therefore has an additional GP map in the system which the mathematical model makes use of. The existence of an additional lower-level GP map enlarges the empirical scope of the paper, and it could be worthwhile to reflect on the implications of this in connection with multiscale biology.

We thank the reviewer for the thoughtful comment. We have added the following text to the discussion:

“Together with additional work⁵⁸, this highlights the importance of multi-scale models in biology. In particular, although there are many models for how biochemical parameters influence higher-level cellular and organ phenotypes, these models rarely connect to genetic variation. Deep mutagenesis of additional biological processes, including those with more complex dynamical behaviour, will provide a more complete view of how mutations impact on phenotypes and fitness.”

What I find most intriguing with the paper is that the system under study shows such complex sign epistasis behaviour. This is a phenomenon that is of considerable significance in biomedicine, production biology and evolutionary biology. And I have to admit that I was a bit surprised that even this simple system can display so much sign epistasis. Sign epistasis is caused by order breaking of the GP map, i.e. the partial order of genotypes breaks with the order of the associated phenotypes such that GP map becomes non-monotonic. Monotonicity is a mathematical term wherefrom we can develop theory while

sign epistasis is just a descriptive term caused by the underlying non-monotonicity. The authors do not qualify this link between monotonicity and sign epistasis. Thus I think it would be a worthwhile exercise to interpret the results in terms of monotonic and non-monotonic features of the GP map revealed by the experimental study. It has been shown, by using a monotonicity measure for many-loci systems, that in a wide class of gene regulatory networks most GP maps are monotonic but that the presence of positive feedback and incoherent feedforward predisposes a network to possess non-monotonic GP map features. As the experimental system under study does neither contain positive feedback loops nor incoherent feedforward motifs, the results certainly bring new insight to the table in terms of how frequent non-monotonic GP maps might arise. I think it would strengthen paper if the authors made this point clear.

We very much agree with the reviewer's points about the origins of sign epistasis from nonmonotonicity and its importance. However, our system does not generate sign epistasis i.e. mutation effects that switch from positive to negative in different genetic backgrounds. Rather, what we observe is that a change in expression can switch the 'sign of the epistasis' itself i.e. switch positive interactions to negative interactions (not positive mutation effects to negative mutation effects).

We have attempted to clarify this in the revised manuscript (result section 'Changes in gene expression reverse the direction of genetic interactions'):

"It is worth noting that these changes are different from sign epistasis which refers to the mutational effects themselves switching from positive to negative in different genetic backgrounds⁴⁹. We did not observe sign epistasis in our experiment or model."

We also now use the term "direction of genetic interaction" instead of "sign of epistasis" to avoid confusion with the widely used term 'sign epistasis'.

Overall, I think this paper is important for providing new data on the complexity of epistasis patterns that exist even in a very simplified setting and which transfers to many domains, it shows how a quite simple mechanistic model can explain much of this complexity, it demonstrates a new mechanism for creating non-monotonic GP maps, and it provides a theoretical-experimental foundation for systematic elucidation of how the GP map changes as a function of mutational changes affecting both expression and structure of regulatory proteins. For these reasons, the results should be of interest to the community and the wider field.

I am confident that the results are reproducible as the description of the experimental and theoretical work is pretty well articulated.

Again, we thank reviewer #1 for the positive evaluation of the manuscript.

Minor comments:

Line 24: There are many aspects connected to the challenge of making genetic predictions. Consider to change the phrasing “To understand the reasons for this” to something more moderate and informative, for example, “we wish to reveal how intricate the GP map might be even in a very simple gene regulatory context.”

We have edited this sentence to read:

“To better understand the plasticity of genetic interactions (epistasis), we combined mutations in a single protein performing a single function (a transcriptional repressor inhibiting a target gene).”

Line 26: The term “genetic interaction” is loaded with different interpretational content depending on which field you come from. The term comes from statistical genetics but is used widely as shorthand in other communities, but with different interpretations. However, the mutations as such do not interact in the sense that they have a reciprocal effect on each other as such. Their interdependence is manifested in the GP map, and we would benefit from using more precise wording as long as we want to develop causal explanatory structures.

We are careful to define how we quantify epistasis in the manuscript:

“We quantified epistasis between pairs of mutations as the difference between the observed and expected phenotypes based on a log additive model⁵⁰.”

Line 59: Skip Why is this? It is redundant.

Deleted in the revised manuscript.

Some headings in the Results section make generic claims that are not original. Consider some more specific wording.

We reworded three headings to be more specific:

- 1) *Mutation effects change non-linearly with a change in expression*
 - *Mutational effects **in CI** change non-linearly with a change in expression*
- 2) *Changing expression alters how mutations interact*
 - *Changing expression alters how mutations **in CI** interact*
- 3) *Non-monotonic expression-phenotype relationships result in ambiguous genetic prediction*
 - *Non-monotonic expression-phenotype relationships result in ambiguous genetic prediction **when more than one mutation is combined***

Skip the first two paragraphs in Discussion. Instead of telling me what you have just told me, use the space to qualify possible shortcomings and outline the generic value of the results. Start with the possible shortcomings to meet the readers' questions up front.

We have reduced the summary of the main findings at the start of the Discussion to one short paragraph, but think that it is important to include this, as is done in very many papers. We moved the paragraph on possible shortcomings immediately after the first summary text.

Line 267: The plasticity of epistasis is not an original finding. There is a rich literature on this so either you contextualise or you skip the paragraph. I suggest the latter.

We skipped the paragraph as suggested by the reviewer. This material is covered in the introduction.

Line 272: There already exist theoretical literature on the propagation of mutational effects across scales and in gene regulatory networks. Place your claim and results in this paragraph in the existing context or skip the paragraph.

We thank the reviewer for his/her suggestion and have cited more of this inspirational work in the discussion.

Reviewer #2 (Remarks to the Author)

Thank you for the opportunity to review “Changes in gene expression predictably shift and switch genetic interactions” by Li et al. This manuscript describes how varying expression level of a gene changes the direction and magnitude of single mutant effects as well as pairwise genetic interactions between mutations. Authors use phage lambda repressor (CI), a well-studied protein and a model for gene regulation, expressed at two different levels (low and high) to test the effect of single and pairwise mutations on its transcriptional activity. They use ‘doped’ oligonucleotide synthesis to randomly introduce mutations into the 59 amino acid helix-turn-held DNA-binding domain of CI and use GFP fluorescence intensity as a readout for the transcriptional activity of CI variants. They use FACS to separate cells into two bins with neutral and intermediate population based on GFP fluorescence and then use deep sequencing to identify the variants in each bin.

Authors then quantify single mutant effects as well as genetic interactions between mutations and generate a model for their prediction by integrating protein folding and protein concentration. They show that changes in gene expression can affect the magnitude and the direction of genetic interactions between mutations.

The idea to investigate the effect of how gene expression could modify the effects of single and pairwise combinations of mutations within a gene of interest is interesting and important. Other studies examine the effect of mutations singly and in combination on protein function by studying a gene of interest at a defined expression level, whereas the novelty of this study is the comparison of these effects between two different gene expression levels. This is an elegant study and the manuscript is well-written and logical. I only have minor suggestions that the authors should consider in a revised manuscript.

We thank reviewer #2 for the positive evaluation and helpful comments. Please see below for the detailed responses to each suggestion.

1. On page 4 authors should explain how 18 single and 4 double mutants were chosen to compare the behaviour of mutants between bulk sequencing and individual testing.

The mutants were chosen as explained in the revised methods section:

“22 genotypes (Supplementary Figure 2c,d and Supplementary Table 2) were selected based on their enrichment scores at both CI concentrations for re-testing in order to cover a wide phenotypic space. In this reference set, we included all mutation types including synonymous, nonsense, missense and also some double mutations.”

In response to reviewer #3 ‘s suggestion, we tested 9 more genotypes as another independent set and verified GFP estimation from the sequencing data (Supplementary Figure 2) and added the description to the method section as well:

“ In order to verify estimated GFP expression levels converted from the enrichment scores based on the reference set of 22 genotypes (see below Data analysis section), we selected 9 additional genotypes (Supplementary Figure 2, Supplementary Table 3) after mapping the enrichment scores to the target gene GFP expression levels. The experiment procedure was the same for the 22 genotypes mentioned above.”

2. On the same page, authors claim that mutations to less similar amino acids were more detrimental and mutations that introduced a negative charge were also more detrimental at DNA contact sites, but it is unclear if this difference is statistically significant since Figure S2C, S2D, S2I and S2J do not show results of any statistical tests.

We have added statistical test results to Figure panels S2C, S2D, S2I and S2J. Specifically, to examine whether there is significant correlation between the Amino acid replacement matrix score (Blosum62) and the variants' GFP intensity, we performed spearman rank correlation test, and reported the Spearman correlation rho, as well as 'Bonferroni' adjusted p-values from multiple comparisons. At both expression levels of CI, Blosum62 scores are best correlated with amino acid substitutions of residues at the core (rho= -0.55 and -0.56 for low and high expression of CI respectively, p-value<1e-6), moderately correlated at the DNA-contacting sites at low expression (rho= -0.26, p-value=0.02) and least at the surface (and not statistically significant at either expression levels of CI). To examine whether changes in the side-chain charge affect the variants GFP intensity, Kruskal-wallis test with a post hoc Dunn's test was performed. Also, the 'Bonferroni' adjusted p-values from multiple comparisons were reported together. At the DNA-contacting sites, influence of introducing negative charge is significantly different (much more detrimental) from substitutions that do not change the charge or introduce positive charge (chi-squared= 15.6, p-value<=0.003). At other sites, changes in side-chain charges do not produce significant differences except at the surface at the low expression, where introducing negative charge is more detrimental than introducing positive charge (chi-squared= 6.3, p-value=0.02).

3. The data presented in Figure S2E and S2F are not convincing since most points are oriented vertically suggesting that delta G folding or delta G binding shows no correlation with GFP intensity and that the correlation is driven by 5 outliers (in case of delta G binding). It may be helpful to separate the plots into solvent-exposed, core and contacting DNA sub-plots otherwise because the points overlap, it is difficult to see a trend. It may also be helpful to bin delta G and use additional complementary methods to calculate delta G folding and binding to make robust conclusions.

We replaced the figure panel with binned deltaGs, as suggested by the referee.

4. The Methods section on 'Sorting' on pg 19 describes 3 gates (near neutral, completely detrimental and intermediate) that were used to sort cell populations.

Authors should explain why they only used near neutral and intermediate populations for deep sequencing.

We added the explanation to the Methods section on 'Sorting' in the revised manuscript:

“Cells from the completely detrimental gate were not further processed for deep sequencing, for the following two reasons: 1) Variants from the detrimental gate were expected to be enriched with insertions or deletions, and stop codons that we do not want in our analysis; 2) amino acid substitutions that are completely detrimental (therefore enriched in the completely detrimental gate) can be deduced based on variants' frequency in the input, near neutral fraction, and intermediate fraction.”

5. In Figure 2, the total number of interactions in 4 different classes does not add up to 100% but to 93%. Authors should account for the difference.

As the reviewer pointed out, there are variants that fall out of these four classes. We now note this in the main text.

“7% of mutants, including mutations partially detrimental at both expression levels, did not fall into either of these four classes.”

It is surprising that there are no points in the top left quadrant of Figure 2D and 2E? There are no CI variants that have no effect at low expression but are detrimental at high expression? Would this subset be observed if higher order interactions were examined or is this a technical limitation of the experiment or the quantification?

Our experiment and quantification method would allow us to detect variants that have no effect at low expression but are detrimental at high expression, if there are any. Variants that are toxic upon overexpression would, for example, fall into this category. However, as pointed out by the reviewer, we did not observe any variants of this type in our dataset, suggesting that they are rare for single and double mutants.

6. The model, which is described in Figure 3, includes protein folding and protein concentration as parameters to predict the effect of mutations. However, it is known that CI easily dimerizes, so it would be prudent to include the effect of mutations on dimerization status/kinetics into the model.

Indeed, CI dimerization is important and mutations affecting dimerization status/kinetics will have phenotypic effects. However, we did not consider mutational effects on dimerization because we only mutated the helix-turn-helix domain of CI. We did not mutate the C terminal domain that contains the dimerization interface. Therefore, in our model, the dimerization energy is kept as the wild type parameter as originally described in the Ackers' model.

7. It would be helpful to discuss higher order genetic interactions and to what extent they are going to be modified by gene expression changes.

This is a very interesting question and one that we hope to address in future work. Non-linear genotype-phenotype maps also generate higher-order genetic interactions (see for example Baeza et al. Cell 2019). We expect these higher order genetic interactions to also be modified by gene expression changes as for pair-wise genetic interactions in our current model.

Additionally, the majority (62%) of interactions does not change due to differences in CI gene expression. Authors should not lose the sight of the stability of genetic interaction networks.

We agree with the reviewer's point that the majority of interactions are stable, if we only consider directional changes in genetic interactions as changes.

Considering all possible combinations of single amino acid substitutions observed in our data set, the model predicts that ~37% of mutation pairs change the direction of genetic interaction (positive interaction at low expression, and negative interaction at high expression). But if we consider magnitude changes of genetic interactions as well, the majority of pair-wise genetic interactions also change (as shown in Figure 4a).

8. Supplementary figures should be called out sequentially so on pg 5 Figure S5A is referenced immediately after Figure S2. It may be prudent to move it up and rename it to Figure S3 to enable easy navigation through the manuscript.

We have reordered the supplementary figures as suggested by the reviewer.

Reviewer #3 (Remarks to the Author):

Summary:

The authors use the phage lambda repressor (CI) as a model to study how the effects of mutations, and their interactions, change across two different expression levels. Having constructed a mutant plasmid library, where every plasmid contains GFP and a mutagenized CI, they use fluorescence as a readout of CI ability to repress GFP. After fluorescence-activated cell sorting into two bins, the mutagenized region is sequenced, and the enrichment of reads in every bin is used to predict the GFP signal of every mutant CI genotype. They then go on to estimate single mutation and interaction effects, and show how the differences in these effects across the two expression levels is predicted by a mathematical model involving folding and binding energies of CI.

Overall evaluation:

I find this manuscript very interesting. The point about genetic interactions emerging as a consequence of a non-linear relationship between traits, such as gene expression and fitness, has been made before. However, the fact that these effects will also depend on the expression “baseline” has to my knowledge not been articulated previously. The authors also use an appealing model system (the phage lambda repressor) to empirically study this phenomenon which is a novel contribution, since this has previously largely been a theoretical argument. Finally, the point about non-monotonic expression-phenotype relationships resulting in ambiguous phenotypic predictions raises many interesting questions. Overall, I think this study makes novel contributions that might be of general relevance for our understanding of genotype-phenotype relationships, and ultimately our ability to predict phenotypes from genotypes. However, I’m not fully convinced that the empirical results presented support the main claims (see detailed comments).

We thank reviewer #3 for the critical evaluation. Please see below for the detailed responses to each suggestion.

Major Comments:

1) My first major concern regards the writing of the manuscript. I found it impossible to understand how the experiment was done based on the results section alone. Many crucial details have been left out, such as the fact that the GFP expression per genotype that the authors use in subsequent analyzes are not directly observed, but indirectly inferred from enrichment of reads in the two sorted bins. The reader has to dig quite deep in the long methods section in order to understand what the authors did, which makes this a very “heavy” read. I would appreciate an extended version of fig1A-C describing all the major steps in the experiment, and a few more details in the results section.

We have added a new supplementary figure (Supplementary Figure 1) that presents the entire experimental data processing procedure up to the prediction of GFP expression levels.

We also added a few more details in the results section:

“From the sequencing read counts, we calculated enrichment scores for each variant in each bin. We then estimated the GFP expression using the enrichment scores from both bins (Figure 1c, Supplementary Figure 1, see Methods).”

2) The analysis in this manuscript involves a rather long chain, where an indirectly inferred metric is used to infer another metric and so on. My biggest concern here is the GFP level per mutant genotype, which most subsequent analyzes depend upon. This is inferred from the amount of enriched reads in the two sorted bins. In order to validate this method, the authors individually quantify the GFP levels of 22 genotypes and compare it to the predicted GFP (Fig1D). Although the prediction looks good in these cases, the 22 genotypes were picked because they were highly reproducible in terms of read enrichment. To me, this does not sound like a fair representation of the mutant library. Would it be possible to perform the same validation with a random sample of mutant genotypes?

As we now show in Supplementary Figure 2b, the variance of the predicted GFP levels of the 22 genotypes is not smaller than the variance of the entire dataset analyzed in the manuscript. The 22 genotypes were selected from the data analyzed in the manuscript (all data points were with standard Error <1), not because their variances were smaller than rest of the data. We apologize for not making it clear in the manuscript of the earlier version.

We added more information as for how we selected the 22 genotypes in the revised manuscript Method:

“22 genotypes (Supplementary Figure 2c,d and Supplementary Table 2) were selected based on their enrichment scores at both CI concentrations for re-testing in order to cover a wide phenotypic space. In this reference set, we included all mutation types including synonymous, nonsense, missense and also some double mutations.”

In response to the reviewer’s comment, in the revised manuscript we have re-tested an additional 9 genotypes (Supplementary Figure 2a, Supplementary Table 3). The estimated GFP signals from the bulk experiment correlate with the individually validated data with Spearman rank correlation of 0.72 (p-value=0.001) compared to rho=0.87 for the originally tested 22 genotypes.

3) The authors compare four different mathematical models and conclude that the “full model”, which allows for non-linearity both in protein folding and regulatory interaction, best fits the data. While this does sound reasonable, the full model should also be most flexible to fit any data since it has more parameters. I’m thus suspicious that the better fit is at least in part due to overfitting. Would it be possible to perform some sort of out of sample prediction, where test and training data are kept separate, to validate these models?

We appreciate the reviewer's concern about overfitting and indeed the full model does have more parameters. However, none of these parameters were fitted by us. As presented in Supplementary Table 8 (copied below), all of the parameters in the full model were taken from the literature except for the expression level of the WT proteins at high and low expression, which we measured in independent experiments.

Supplementary Table 8. Parameters for CI model, from the literature

K_a	5×10^7	Ackers, 1982
[OR]	10^{-9} mole	Ackers, 1982
ΔG_1	-11.7 kcal	Ackers, 1982
ΔG_2	-10.1 kcal	Ackers, 1982
ΔG_3	-10.1 kcal	Ackers, 1982
ΔG_{co}	-2 kcal	Ackers, 1982
CI fraction folded	0.993	Huang, 1995

Minor Comments:

1) The authors point out that double mutations with high expected target gene expression tended to interact positively and vice versa (lines 139-143). Isn't this expected per definition? They define positive interactions as less target gene expression than expected (Fig 2F). So this should by definition happen when the expectation is high right?

Thank you for pointing this out. Indeed, it is expected that high-expected target gene expression tended to interact positively. In response to this comment, we rewrote the sentence as below:

“As expected, double mutants with high expected target gene expression tended to interact positively at both low and high expression. On the other hand, double mutants with intermediate expected outcomes had stronger negative interactions at low expression, and double mutants with low expected target gene expression had stronger negative interactions at high expression (Supplementary Figure 4).”

2) Why did the authors not sequence genotypes in the “detrimental gate” after sorting? Economic reasons? Wouldn't an enrichment score based on all three bins, rather than just two, give a better prediction of GFP levels?

We agree that enrichment scores from more bins would improve the prediction of GFP levels and our choice of bins was indeed driven by economic considerations.

Reviewers' Comments:

Reviewer #1:

Remarks to the Author:

I am happy with the revisions of the ms.

Just one note to the reply:

"We very much agree with the reviewer's points about the origins of sign epistasis from nonmonotonicity and its importance. However, our system does not generate sign epistasis i.e. mutation effects that switch from positive to negative in different genetic backgrounds. Rather, what we observe is that a change in expression can switch the 'sign of the epistasis' itself i.e. switch positive interactions to negative interactions (not positive mutation effects to negative mutation effects)."

I may have misunderstood this, but as the change in expression mimics the effect of a mutation in the promoter or enhancer region of your gene that in principle could have regulated the expression, I would think that the system as such is in principle capable of producing sign epistasis. Thus, you are justified in claiming that if the system includes the above type of mutations, sign epistasis will result. It just enlarges the empirical scope of the paper, but the phrasing in the revised ms is ok.

Response to referees' comments

REVIEWERS' COMMENTS:

Reviewer #1 (Remarks to the Author):

I am happy with the revisions of the ms.

Just one note to the reply:

"We very much agree with the reviewer's points about the origins of sign epistasis from nonmonotonicity and its importance. However, our system does not generate sign epistasis i.e. mutation effects that switch from positive to negative in different genetic backgrounds. Rather, what we observe is that a change in expression can switch the 'sign of the epistasis' itself i.e. switch positive interactions to negative interactions (not positive mutation effects to negative mutation effects)."

I may have misunderstood this, but as the change in expression mimics the effect of a mutation in the promoter or enhancer region of your gene that in principle could have regulated the expression, I would think that the system as such is in principle capable of producing sign epistasis. Thus, you are justified in claiming that if the system includes the above type of mutations, sign epistasis will result. It just enlarges the empirical scope of the paper, but the phrasing in the revised ms is ok.

We thank Dr.Omhold for the positive evaluation and the thoughtful discussions.

Reviewer #2 (Remarks to the Author):

The authors have sufficiently addressed most comments.

We thank Reviewer #2.

Reviewer #3 (Remarks to the Author):

The authors have addressed all of my previous concerns. I find the revised version of the manuscript significantly improved.

We thank Dr. Forsberg for the positive evaluation of our revised manuscript.

1) The new figure S1 greatly clarifies the experimental pipeline.

We thank the reviewer for suggesting to include the experimental pipeline.

2) I thank the authors for this clarification, and for validating the estimated GFP signals in 9 additional genotypes. I believe that figures 1d and S2a,b makes a compelling validation of this method to infer GFP signals.

Again, we thank the reviewer.

3) I apologize for my misunderstanding. The fact that these parameters were taken from the literature, rather than fitted from the data, makes the results even more interesting.

We thank the reviewer for raising the issue, therefore, we had the opportunity to make it clearer in the text.

Finally, I'd like to thank the authors for this interesting work. I'm intrigued by the observation that interactions that arise because of non-linear relationships between phenotypes are dependent on the "baseline" of the lower level phenotype. This might be one reason why genetic interactions are abundantly observed in well controlled model systems such as this, but are hard to detect in for instance GWAS. In a GWAS where the genetic background and environment varies between individuals, it is likely that individuals will have different baselines in gene expression and other lower level phenotypes. The magnitude and sign of any genetic interactions, in terms of the higher level observed phenotype, might therefore be unique to every individual, making it hard or impossible to infer said interactions on the population level. This obviously also has great implications for phenotypic prediction.

We thank Dr. Forsberg for thoughtful comment on our work and recognizing the value of our findings.